# Large-scale neural dynamics in a shared low-dimensional state space reflect cognitive and attentional dynamics

Hayoung Song[1]*, Won Mok Shim[2,3,4]*[†], Monica D Rosenberg[1,5]*[†]

[1]Department of Psychology, University of Chicago, Chicago, United States; [2]Department of Biomedical Engineering, Sungkyunkwan University, Suwon, Republic of Korea; [3]Center for Neuroscience Imaging Research, Suwon, Republic of Korea; [4]Department of Intelligent Precision Healthcare Convergence, Sungkyunkwan University, Suwon, Republic of Korea; [5]Neuroscience Institute, University of Chicago, Chicago, United States

**Abstract** Cognition and attention arise from the adaptive coordination of neural systems in response to external and internal demands. The low-dimensional latent subspace that underlies large-scale neural dynamics and the relationships of these dynamics to cognitive and attentional states, however, are unknown. We conducted functional magnetic resonance imaging as human participants performed attention tasks, watched comedy sitcom episodes and an educational documentary, and rested. Whole-brain dynamics traversed a common set of latent states that spanned canonical gradients of functional brain organization, with global desynchronization among functional networks modulating state transitions. Neural state dynamics were synchronized across people during engaging movie watching and aligned to narrative event structures. Neural state dynamics reflected attention fluctuations such that different states indicated engaged attention in task and naturalistic contexts, whereas a common state indicated attention lapses in both contexts. Together, these results demonstrate that traversals along large-scale gradients of human brain organization reflect cognitive and attentional dynamics.

## Editor's evaluation

This valuable study examines the distribution of four states of brain activity across a variety of cognitive conditions, linking systems neuroscience with cognition and behavior. The work is convincing, using null models and replication in independent datasets to support their findings.

## Introduction

A central goal in cognitive neuroscience is understanding how cognition arises from the dynamic interplay of neural systems. To understand how interactions occur at the level of large-scale functional systems, studies have characterized neural dynamics as a trajectory in a latent state space where each dimension corresponds to the activity of a functional network (*Breakspear, 2017*; *Gu et al., 2015*; *John et al., 2022*; *Shine et al., 2019a*). This dynamical systems approach revealed two major insights. First, neural dynamics operate on a low-dimensional manifold. That is, neural dynamics can be captured by a small number of independent latent components due to covariation of neural activity within a system (*Cunningham and Yu, 2014*; *Shine et al., 2019b*). Second, neural activity does not just continuously flow along a manifold, but rather systematically transitions between recurring latent 'states,' or hidden clusters, within the state space (*Baker et al., 2014*; *Chen et al., 2016*;

*For correspondence:
hyssong@uchicago.edu (HS);
wonmokshim@skku.edu (WMS);
mdrosenberg@uchicago.edu
(MDR)

[†]These authors contributed
equally to this work

Competing interest: The authors declare that no competing interests exist.

*Vidaurre et al., 2018*; *Vidaurre et al., 2017*). Initial work used resting-state neuroimaging (*Allen et al., 2014*; *Betzel et al., 2016*; *Bolt et al., 2022*; *Liu and Duyn, 2013*; *Yousefi and Keilholz, 2021*; *Zalesky et al., 2014*; *Zhang et al., 2019*) and data simulations (*Deco et al., 2017*; *Deco et al., 2015*; *Friston, 1997*) to describe dynamic interactions among brain regions in terms of systematic transitions between brain states.

Less is known about how our mental states—which constantly ebb and flow over time—arise from these brain state transitions. Recent work in human neuroimaging suggests that brain state changes reflect cognitive and attentional state changes in specific contexts (*Gao et al., 2021*; *Shine et al., 2019a*). For example, work has identified neural states during a sustained attention task (*Yamashita et al., 2021*) or a working memory task (*Cornblath et al., 2020*; *Taghia et al., 2018*). Dataset-specific latent states occurred during different task blocks as well as moments of successful and unsuccessful behavioral performance. Another line of work identified latent states during naturalistic movie watching and demonstrated how neural dynamics relate to contents of the movies (*van der Meer et al., 2020*) or participants' ongoing comprehension states (*Song et al., 2021b*). An open question is whether the *same* latent states underlie cognitive states across all contexts. For example, does the same brain state underlie successful attention task performance and engaged movie watching? If brain activity traverses a common set of latent states in different contexts, to what extent do the functional roles of these states also generalize?

*Shine et al., 2019a* demonstrated that neural activity traverse a common low-dimensional manifold across seven cognitive tasks. The dynamics within this common manifold were aligned to exogenous task blocks and related to individual differences in cognitive traits. Here we expand on this work by probing a common set of latent states that explain neural dynamics during task, rest, and naturalistic contexts in five independent datasets. We also identify the nature of this shared latent manifold by relating it to the canonical gradients of functional brain connectome (*Margulies et al., 2016*). Finally, we relate neural state dynamics to *ongoing* changes in cognitive and attentional states to probe how neural dynamics are adaptively modulated from stimulus-driven and internal state changes.

We collected human fMRI data, the *SitcOm, Nature documentary, Gradual-onset continuous performance task (SONG) neuroimaging dataset*, as 27 participants rested, performed attention tasks, and watched movies. We characterized latent state dynamics that underlie large-scale brain activity in these contexts and related them to changes in cognition and attention measured with dense behavioral sampling. Each participant performed seven fMRI runs over 2 d: two eye-fixated resting-state runs, two gradual-onset continuous performance task (gradCPT) runs with either face or scene images, two runs of comedy sitcom watching, and a single run of educational documentary watching. The gradCPT measures fluctuations of sustained attention over time (*Esterman et al., 2013*; *Rosenberg et al., 2013*) as participants respond to images (every 1 s) from a frequent category (90% of trials) and inhibit response to images from an infrequent category (10%). The sitcom episodes were the first and second episodes of a South Korean comedy sitcom, *High Kick Through the Roof*. The educational documentary described the geography and history of Korean rivers.

## Functional brain activity transitions between states in a common latent manifold

### Large-scale neural activity transitions between discrete latent states

To infer latent state dynamics, we fit a hidden Markov model (HMM) to probabilistically infer a sequence of discrete latent states from observed fMRI activity (*Rabiner and Juang, 1986*). The observed variables here were the BOLD signal time series from 25 parcels in a whole-brain parcellation of the cortex (17 functional networks) (*Yeo et al., 2011*) and subcortex (8 regions) (*Tian et al., 2020*) sampled at a 1 s TR resolution (*Figure 1A*, left). Parcel time courses were *z*-normalized within each participant and concatenated across all fMRI runs from all participants. The model inferred two parameters from these time series: the emission probability and the transition probability (see 'Materials and methods'). We assumed that the emission probability of the observed variables follows a mixture Gaussian characterized by the mean and covariance of the 25 parcels in each latent state (*Figure 1A*, right). The inferred parameters of the model were used to decode latent state sequences. Four was chosen as the number of latent states ($K = 4$) based on the optimal model fit to the data when tested with leave-one-subject-out cross-validation (chosen among $K$ of 2–10; *Figure 1—figure supplement 1*).

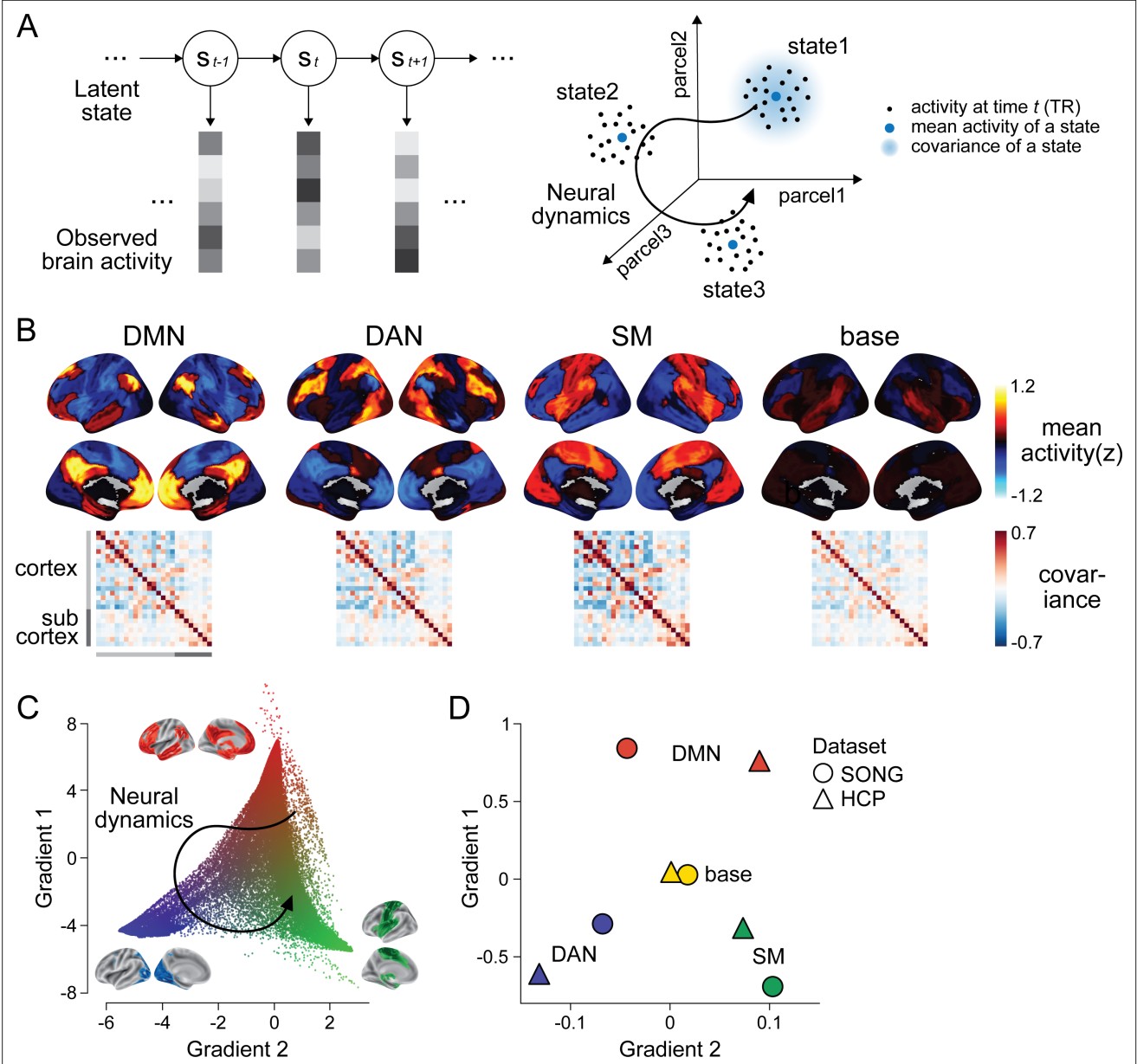

**Figure 1.** Latent state space of the large-scale neural dynamics. (**A**) Schematic illustration of the hidden Markov model (HMM) inference. (Left) The HMM infers a discrete latent state sequence from the observed 25-parcel fMRI time series. (Right) The fMRI time course can be visualized as a trajectory within a 25-dimensional space, where black dots indicate activity at each moment in time. The HMM probabilistically infers discrete latent clusters within the space, such that each state can be characterized by the mean activity (blue dots) and covariance (blue shaded area) of the 25 parcels. (**A**) has been adapted from Figure 1A from *Cornblath et al., 2020*. (**B**) Four latent states inferred by the HMM fits to the SONG dataset. Mean activity (top) and pairwise covariance (bottom) of the 25 parcels' time series is shown for each state. See *Figure 1—figure supplement 6* for replication with the Human Connectome Project (HCP) dataset. (**C**) Conceptualizing low-dimensional gradients of the functional brain connectome as a latent manifold of large-scale neural dynamics. Each dot corresponds to a cortical or subcortical voxel situated in gradient space. The colors of the brain surfaces (inset) indicate voxels with positive or negative gradient values with respect to the nearby axes. Data and visualizations are adopted from *Margulies et al., 2016*. (**D**) Latent neural states situated in gradient space. Positions in space reflect the mean element-wise product of the gradient values of the 25 parcels and mean activity patterns of each HMM state inferred from the SONG (circles) and HCP (triangles) datasets.

The online version of this article includes the following figure supplement(s) for figure 1:

**Figure supplement 1.** The choice of the number of states (K) in latent state inference.

**Figure supplement 2.** Latent state inference conducted separately to each condition of the SONG dataset.

*Figure 1 continued on next page*

*Figure 1 continued*

**Figure supplement 3.** Examples of the null latent states derived from the hidden Markov models (HMMs) conducted on the surrogate fMRI time series of the SONG dataset (1000 iterations).

**Figure supplement 4.** Latent state inference using a different whole-brain parcellation scheme.

**Figure supplement 5.** The inferred latent states and their dynamics are robust to the choice of the fMRI preprocessing approach.

**Figure supplement 6.** Latent state inference on the Human Connectome Project (HCP) dataset.

**Figure supplement 7.** The position in predefined gradient space at every time point grouped by hidden Markov model (HMM) latent state.

**Figure supplement 8.** Comparisons between predefined and data-specific gradients.

*Figure 1B* illustrates the four latent neural states inferred by the HMM in the SONG dataset (see *Figure 1—figure supplement 2* for condition-specific latent states). We labeled three states the default mode network (DMN), dorsal attention network (DAN), and somatosensory motor (SM) states based on high activation of these canonical brain networks (*Yeo et al., 2011*). (Note that these state labels are only applied for convenience. Each state is characterized by whole-brain *patterns* of activation, deactivation, and covariance, rather than simply corresponding to activation of the named network.) The fourth state was labeled the 'base' state because activity was close to baseline ($z = 0$) and covariance strength (i.e., the sum of the absolute covariance weights of the edges) was comparatively low during this state. The SM state, on the other hand, exhibited the highest covariance strength, whereas the covariance strengths of the DMN and DAN states were comparable. Compared to null latent states derived from surrogate fMRI time series, the four states exhibited activity patterns more similar to large-scale functional systems (*Buckner et al., 2008*; *Corbetta and Shulman, 2002*; *Fox et al., 2005*; *Smith et al., 2009*) and significantly higher covariance strength (see *Figure 1—figure supplement 3* for examples of null latent states). These states were replicated with 250 regions of interest (ROIs) consisting of 200 cortical (*Schaefer et al., 2018*) and 50 subcortical regions (*Tian et al., 2020*), albeit with a caveat that the HMM provides a poorer fit to the higher-dimensional time series (*Figure 1—figure supplement 4*). Neural state inference was robust to the choice of $K$ (*Figure 1—figure supplement 1*) and the fMRI preprocessing pipeline (*Figure 1—figure supplement 5*) and consistent when conducted on two groups of randomly split-half participants (Pearson's correlations between the two groups' latent state activation patterns: DMN: 0.791; DAN: 0.838; SM: 0.944; base: 0.837).

To validate that these states are not just specific to the SONG dataset, we analyzed fMRI data from the Human Connectome Project (HCP; N = 119) (*Van Essen et al., 2013*) collected during rest, seven block-designed tasks—the emotion processing, gambling, language, motor, relational processing, social cognition, and working memory tasks (*Barch et al., 2013*)—and movie watching (*Finn and Bandettini, 2021*). The same HMM inference was conducted independently on the HCP dataset using $K = 4$ (*Figure 1—figure supplement 6*). HCP states closely mirrored the DMN, DAN, SM, and base states (Pearson's correlations between activity patterns of SONG- and HCP-defined states: DMN: 0.831; DAN: 0.814; SM: 0.865; base: 0.399). Thus, the latent states are reliable and generalize across independent datasets.

## Latent state dynamics span low-dimensional gradients of the functional brain connectome

The HMM results demonstrate that large-scale neural dynamics in diverse cognitive contexts (tasks, rest, and movie watching) are captured by a small number of latent states. Intriguingly, the DMN, DAN, and SM systems that contribute to these states tile the principal gradients of large-scale functional organization. In a seminal paper, *Margulies et al., 2016* applied a nonlinear dimensionality reduction algorithm to capture the main axes of variance in the resting-state static functional connectome of 820 individuals. They found that the primary gradient dissociated unimodal (visual and SM regions) from transmodal (DMN) systems. The secondary gradient fell within the unimodal end of the primary gradient, dissociating the visual processing from the SM systems. These gradients, argued to be an 'intrinsic coordinate system' of the human brain (*Huntenburg et al., 2018*), reflect variations in brain structure (*Huntenburg et al., 2017*; *Paquola et al., 2019*; *Vázquez-Rodríguez et al., 2019*), gene expressions (*Burt et al., 2018*), and information processing (*Huntenburg et al., 2018*).

We hypothesized that the spatial gradients reported by *Margulies et al., 2016* act as a low-dimensional manifold over which large-scale dynamics operate (*Bolt et al., 2022*; *Brown et al., 2021*; *Karapanagiotidis et al., 2020*; *Turnbull et al., 2020*), such that traversals within this manifold explain large variance in neural dynamics and, consequently, cognition and behavior (*Figure 1C*). To test this idea, we situated the mean activity values of the four latent states along the gradients defined by *Margulies et al., 2016* (see 'Materials and methods'). The brain states tiled the two-dimensional gradient space with the base state at the center (*Figure 1D*, *Figure 1—figure supplement 7*). The Euclidean distances between these four states were maximized in the two-dimensional gradient space compared to a chance where the four states were inferred from circular-shifted time series (p<0.001). For the SONG dataset, the DMN and SM states fell at more extreme positions on the primary gradient than expected by chance (both FDR-p-values=0.004; DAN and SM states, FDR-p values=0.171). For the HCP dataset, the DMN and DAN states fell at more extreme positions on the primary gradient (both FDR-p values=0.004; SM and base states, FDR-p values=0.076). No state was consistently found at the extremes of the secondary gradient (all FDR-p values>0.021).

We asked whether the predefined gradients explain as much variance in neural dynamics as latent subspace optimized for the SONG dataset. To do so, we applied the same nonlinear dimensionality reduction algorithm to the SONG dataset's ROI time series. Of note, the SONG dataset includes 18.95% rest, 15.07% task, and 65.98% movie-watching data, whereas the data used by *Margulies et al., 2016* was 100% rest. Despite these differences, the SONG-specific gradients closely resembled the predefined gradients, with Pearson's correlations observed for the first ($r = 0.876$) and second ($r = 0.877$) gradient embeddings (*Figure 1—figure supplement 8*). Gradients identified with the HCP data also recapitulated Margulies et al.'s (2016) first ($r = 0.880$) and second ($r = 0.871$) gradients. We restricted our analysis to the first two gradients because the two gradients together explained roughly 50% of the variance of the functional brain connectome (SONG: 46.94%; HCP: 52.08%), and the explained variance dropped drastically from the third gradients (more than 1/3 drop compared to the second gradients). The degrees to which the first two predefined gradients explained whole-brain fMRI time series (SONG: $r^2 = 0.097$; HCP: 0.084) were comparable to the amount of variance explained by the first two data-specific gradients (SONG: $r^2 = 0.100$; HCP: 0.086; *Figure 1—figure supplement 8*). Thus, the low-dimensional manifold captured by Margulies et al.'s (2016) gradients is highly replicable, explaining brain activity dynamics as well as data-specific gradients, and is largely shared across contexts and datasets. This suggests that the state space of whole-brain *dynamics* closely recapitulates low-dimensional gradients of the *static* functional brain connectome.

## Transient global desynchrony precedes neural state transitions

Neural state transitions can be construed as traversals in a low-dimensional space whose axes are defined by principal gradients of functional brain organization. When and how do these neural state transitions occur? What indicates that the system is likely to transition from one state to another?

We predicted that neural state transitions are related to changes in interactions between functional networks. To test this account, we computed cofluctuation between all pairs of parcels at every TR (1 s). Cofluctuation operationalizes the time-resolved interaction of two regions as an absolute element-wise product of their activity at every time step after *z*-normalization of their time series (*Faskowitz et al., 2020*; *Sporns et al., 2021*; *Zamani Esfahlani et al., 2020*). We time-aligned cofluctuation values to moments of neural state transitions estimated from the HMM (*Figure 2A*). A decrease in cofluctuation prior to the neural state transitions (at time *t-1*) was observed for every pair of cortico-cortical networks ($z = 645.75$, FDR-p=0.001). Cortico-subcortical pairs ($z = 424.05$, FDR-p=0.001) and subcortico-subcortical connections ($z = 64.85$, FDR-p=0.037) also showed decreased cofluctuation before state transitions, although the effects were less pronounced, especially for subcortico-subcortical connections (paired Wilcoxon signed-rank tests comparing the degrees of cofluctuation change, FDR-p-values<0.001). Results were replicated with the 250-ROI parcellation scheme as well as with the HCP dataset (*Figure 2—figure supplement 1*). Furthermore, repeating this analysis with null HMMs on circular-shifted time series suggests that the effect is not simply a by-product of the chosen computational model (*Figure 2—figure supplement 2*). These results are consistent with prior empirical findings that desynchronization, a 'transient excursion away from the synchronized manifold' (*Breakspear, 2002*), allows the brain to switch flexibly between states (*Deco et al., 2017*; *Harris and Thiele, 2011*; *Pedersen et al., 2018*; *Roberts et al., 2019*).

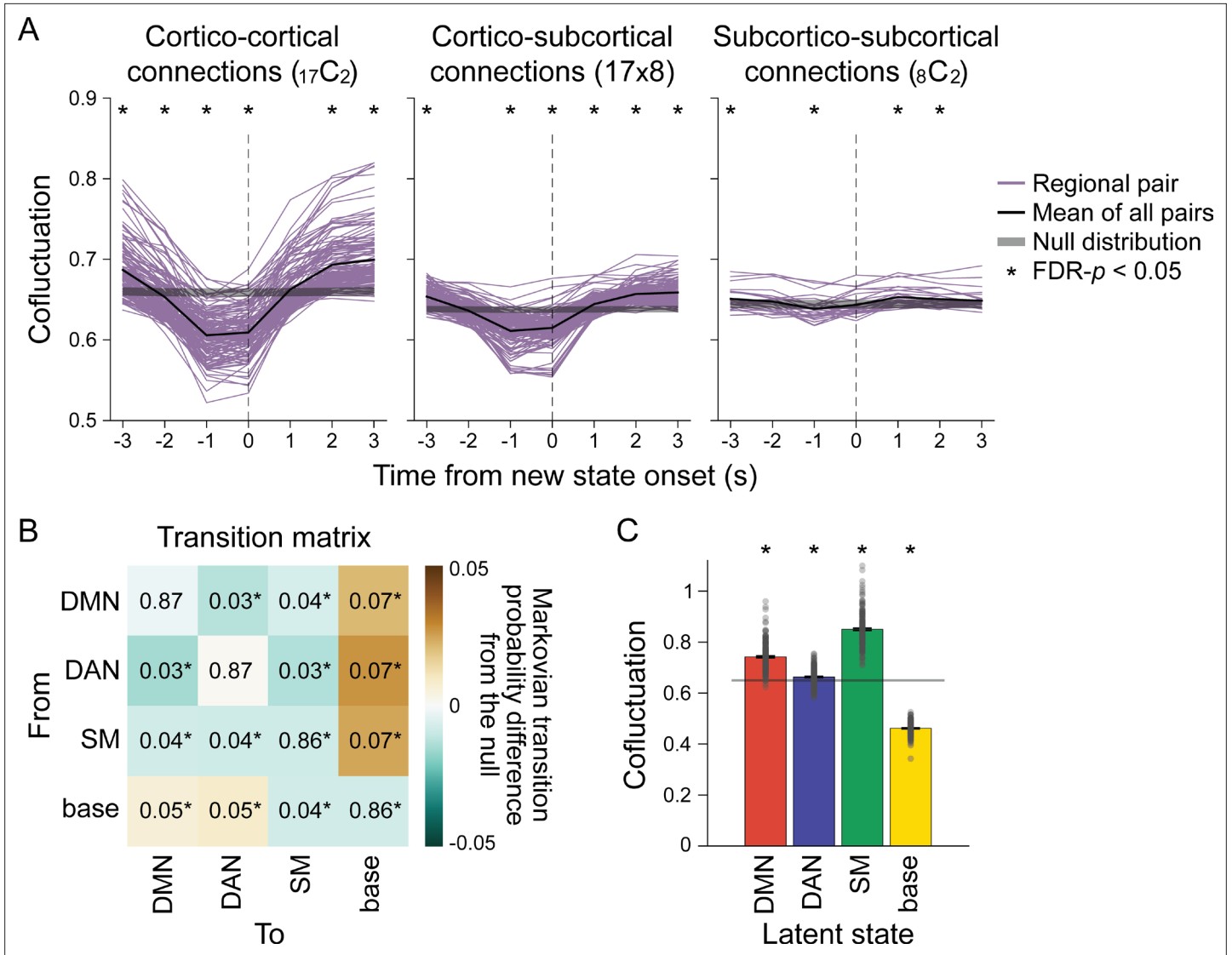

**Figure 2.** Neural state transitions. (**A**) Changes in cofluctuation of the parcel pairs, time-aligned to hidden Markov model (HMM)-derived neural state transitions. State transitions occur between time *t*-1 and *t*. Purple lines indicate the mean cofluctuation of cortico-cortical (left), cortico-subcortical (middle), and subcortico-subcortical (right) parcel pairs across fMRI runs and participants, and the thick black line indicates the mean of these pairs. The shaded gray area indicates the range of the null distribution (mean ± 1.96 × standard deviation), in which the moments of state transitions were randomly shuffled (asterisks indicate FDR-p<0.05). (**B**) Transition matrix indicating the first-order Markovian transition probability from one state (row) to the next (column), averaged across all participants' all fMRI runs. The values indicate transition probabilities, such that values in each row sums to 1. The colors indicate differences from the mean of the null distribution where the HMMs were conducted on the circular-shifted time series. (**C**) Mean degrees of global cofluctuation at moments of latent neural state occurrence. The measurements at each time point were averaged within participant based on latent state identification, and then averaged across participants. The bar graph indicates the mean of all participants' all fMRI runs. The error bars indicate standard error of the mean (SEM). Gray dots indicate individual data points (7 runs of 27 participants). The shaded gray area indicates the range of the null distribution, in which the analyses were conducted on the circular-shifted latent state sequence. See *Figure 2—figure supplement 1* for replication with the Human Connectome Project (HCP) dataset.

The online version of this article includes the following figure supplement(s) for figure 2:

**Figure supplement 1.** Neural state transitions of the Human Connectome Project (HCP) dataset.

**Figure supplement 2.** Cofluctuations of all pairs of 25 parcels of the (**A**) SONG and (**B**) Human Connectome Project (HCP) datasets, time-aligned to the hidden Markov model (HMM)-derived neural state transitions, compared to a null distribution that was generated differently than *Figure 2A*.

**Figure supplement 3.** Transition matrix of the (**A**) SONG and (**B**) Human Connectome Project (HCP) datasets indicating transition probability from one state (row) to the next (column), such that values in each column sums to 1.

**Figure supplement 4.** Mean head motion (framewise displacement [FD]) at moments of latent neural state occurrence in the (**A**) SONG and (**B**) Human Connectome Project (HCP) datasets.

## The base state acts as a flexible hub in neural state transitions

To further address how neural state transitions occur, we analyzed the HMM's transition matrix, which indicates the probability of a state at time *t-1* transitioning to another state or remaining the same at time *t*. The probability of remaining in the same state was dominant (>85%), whereas the probability of transitioning to a different state was less than 8% (*Figure 2B*, *Figure 2—figure supplement 3*). To investigate whether certain state transitions occurred more often than expected by chance, we compared the transition matrix to a null distribution where the HMM was conducted on circular-shifted fMRI time series. The DMN, DAN, and SM states were more likely to transition to and from the base state and less likely to transition to and from one another than would be expected by chance (*Figure 2B*, *Figure 2—figure supplement 3*; FDR-p-values<0.05). The result suggests that the base state acts as a hub in neural state transitions, replicating a past finding of the base state as a transitional hub in resting-state fMRI data (*Chen et al., 2016*).

Given that global desynchrony indicates moments of neural state transitions (*Figure 2A*), we used this measure to validate the role of the base state as a 'transition-prone' state. Cofluctuation between every pair of parcels was computed at every TR, which was averaged across parcel pairs to represent a time-resolved measure of global cofluctuation (*Figure 2C*). When comparing the degree of global cofluctuation across the four latent states, we found that the base state exhibited the lowest degree of global cofluctuation (paired *t*-tests comparing cofluctuation in base state vs. DMN, DAN, and SM states, SONG: $t(187) > 61$, HCP: $t(3093) > 170$, FDR-p-values<0.001), which was significantly below chance (FDR-p-values<0.001). This suggests that the base state was the most desynchronized state among the four, potentially operating as a transition-prone state. Low global synchrony during the base state was not driven by spurious head motion (*Figure 2—figure supplement 4*). Thus, the base state, situated at the center of the gradient space, is a flexible 'hub' state with a high degree of freedom to transition to other functionally specialized states.

## Neural state dynamics are modulated by ongoing cognitive and attentional states
### Latent state dynamics differ across contexts and are synchronized during movie watching

We identified four latent states that recur during rest, task performance, and movie watching. Although the latent manifold of neural trajectories may be shared across contexts, latent states may be occupied to different degrees across contexts. For example, one state may occur more frequently in one context but not in others. We asked whether the pattern with which brain activity 'visits' the four states differed across contexts.

We used the HMM to infer the latent state sequence of each fMRI run (*Figure 3A*) and summarized the fractional occupancy of each state (i.e., proportion of time that a state occurred) (*Figure 3B*; see *Figure 3—figure supplement 1* for dwell time distributions). All four states occurred in all fMRI runs, with no state occurring on more than 50% of time points in a run. Thus, these states are common across contexts rather than specific to one context. Fractional occupancy, however, differed across rest, task, and naturalistic contexts, with strikingly similar values between runs of similar contexts (e.g., rest runs 1 and 2). In contrast to the similar fractional occupancy values of the two sitcom-episode runs, fractional occupancy in the documentary-watching condition differed despite the fact that it also involved watching an audiovisual stimulus. During the documentary, the base state occurred less frequently, whereas the SM state occurred more frequently than during the sitcom episodes.

Latent state dynamics were synchronized across participants watching the comedy sitcom episodes (mean pairwise participant similarity: episode 1: 40.81 ± 3.84%, FDR-p=0.001; episode 2: 40.79 ± 3.27%, FDR-p=0.001; paired comparisons, non-parametric p=0.063; *Figure 3C*). Less synchrony was observed between participants watching the educational documentary (30.39 ± 3.38 %, FDR-p=0.001; paired comparisons with the two sitcom episodes, both p<0.001). No significant synchrony was observed during the resting-state runs (run 1: 25.81 ± 4.00 %, FDR-p=0.230; run 2: 25.84 ± 4.08 %, FDR-p=0.183).

These results were replicated when we applied the SONG-trained HMM to decode latent sequences of the three independent datasets (*Figure 3—figure supplement 2*). The four neural states occurred

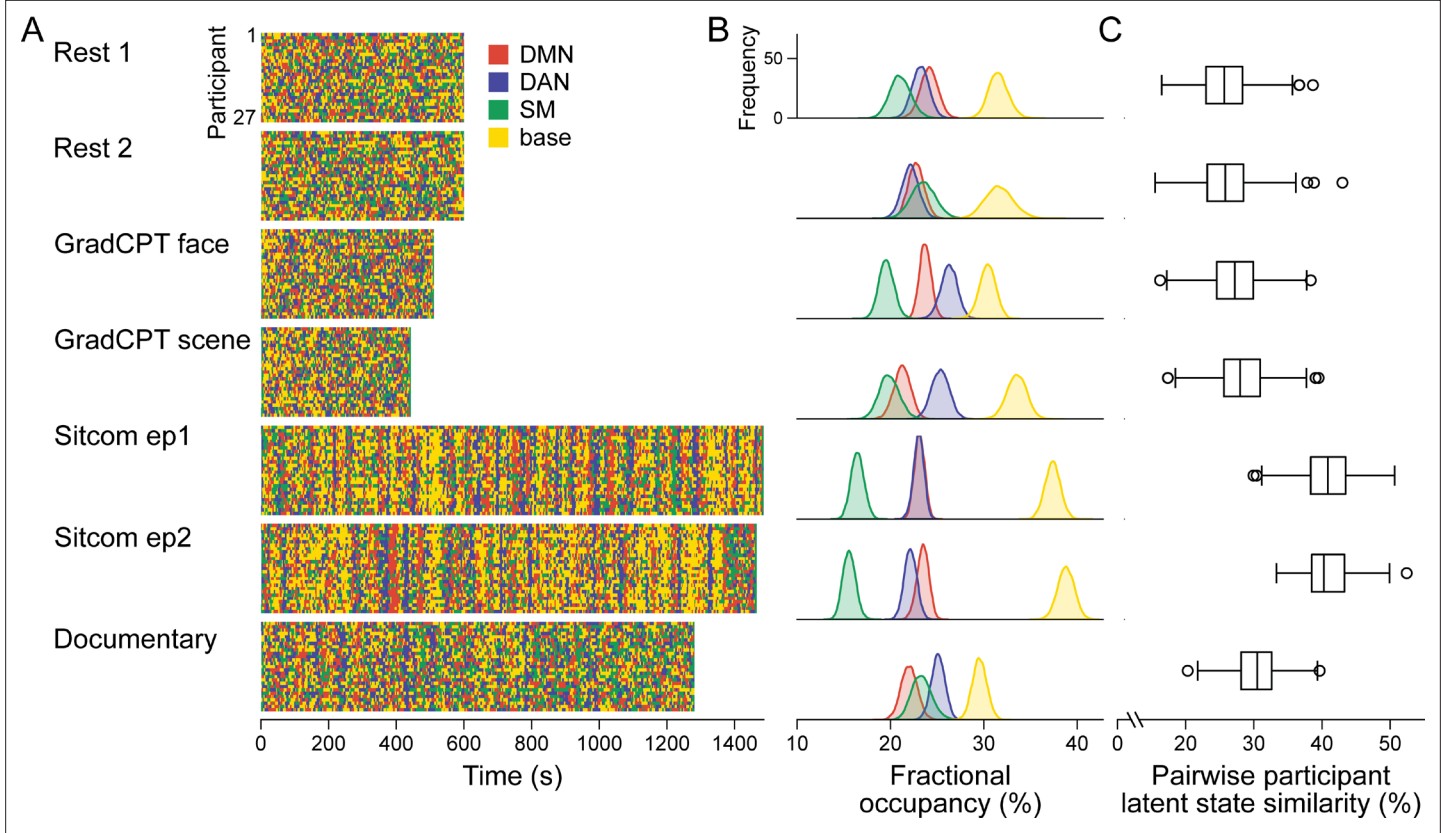

**Figure 3.** Latent neural state dynamics in the seven fMRI runs. (**A**) Latent state dynamics inferred by the hidden Markov model (HMM) for all participants. Colors indicate the state that occurred at each time point. (**B**) Fractional occupancy of the neural states in each run. Fractional occupancy was calculated for each individual as the ratio of the number of time points at which a neural state occurred over the total number of time points in the run. Distributions indicate bootstrapped mean of the fractional occupancies of all participants. The chance level is at 25%. (**C**) Synchrony of latent state sequences across participants. For each pair of participants, sequence similarity was calculated as the ratio of the number of time points when the neural state was the same over the total number of time points in the run. Box and whisker plots show the median, quartiles, and range of the similarity distribution.

The online version of this article includes the following figure supplement(s) for figure 3:

**Figure supplement 1.** Dwell times of the latent neural states, measured as the duration (s) for which a neural state continuously persisted before transitioning to a different state.

**Figure supplement 2.** Inferred neural state dynamics of the external datasets from the hidden Markov model (HMM) trained on the SONG dataset.

in every run of every dataset tested, with maximal fractional occupancies all below 50%. Intersubject synchrony of the latent state sequence was high during movie watching and story listening but at chance during rest. Together the results validate that neural states identified from the SONG dataset generalize not only across contexts but also to independent datasets.

Prior studies reported that regional activity (*Hasson et al., 2004*; *Nastase et al., 2019*) and functional connectivity (*Betzel et al., 2020*; *Chang et al., 2022*; *Simony et al., 2016*) are synchronized across individuals during movie watching and story listening, and that attentional engagement modulates the degree of intersubject synchrony (*Dmochowski et al., 2012*; *Ki et al., 2016*; *Song et al., 2021a*). Our results indicate that the intersubject synchrony occurs not only at regional and pairwise regional scales, but also at a global scale via interactions of functional networks. Furthermore, stronger entrainment to the stimulus during sitcom episodes compared to documentary-watching condition suggests that overall attentional engagement may mediate the degree of large-scale synchrony (mean reports on overall engagement from a scale of 1 [not at all engaging] to 9 [completely engaging]: sitcom episode1: 6.78 ± 1.05, episode2: 6.93 ± 1.41, documentary: 3.59 ± 1.21). Indeed, demonstrating a relationship between neural state dynamics and narrative engagement, participant pairs that exhibited similar engagement dynamics showed similar neural state dynamics (sitcom episode

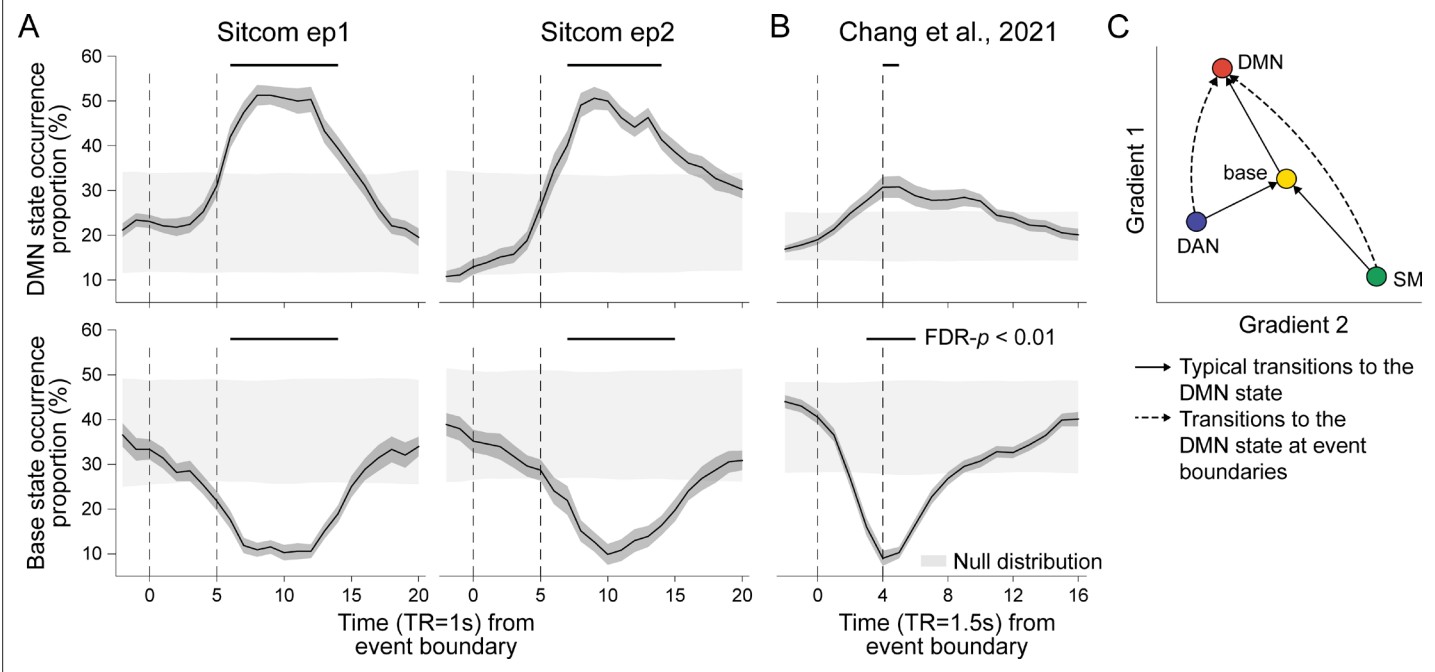

**Figure 4.** Neural state occurrence and transitions at narrative event boundaries. (**A**) The proportion of the default mode network (DMN) (top) and base state (bottom) occurrences time-aligned to narrative event boundaries of sitcom episodes 1 (left) and 2 (right). State occurrence at time points relative to the event boundaries per stimulus was computed within participant and then averaged across participants. The dark gray shaded areas around the thick black line indicate SEM. The dashed lines at $t = 0$ indicate moments of new event onset and the lines at $t = 5$ account for hemodynamic response delay of the fMRI. The light gray shaded areas show the range of the null distribution in which boundary indices were circular-shifted (mean ± 1.96 × standard deviation), and the black lines on top of the graphs indicate statistically significant moments compared to chance (FDR-p<0.01). (**B**) The proportion of the DMN (top) and base state (bottom) occurrence time-aligned to narrative event boundaries of audio narrative. Latent state dynamics were inferred based on the hidden Markov model (HMM) trained on the SONG dataset. Lines at $t = 4$ account for hemodynamic response delay. (**C**) Schematic transitions to the DMN state at narrative event boundaries (dashed lines) compared to the normal trajectory which passes through the base state (solid lines). See *Figure 4—figure supplement 2* for results of statistical analysis.

The online version of this article includes the following figure supplement(s) for figure 4:

**Figure supplement 1.** Neural state occurrence and hippocampal BOLD activity time-aligned to narrative event boundaries.

**Figure supplement 2.** Transitions to the default mode network (DMN) state at narrative event boundaries.

1: Spearman's $r = 0.274$, FDR-p=0.005; episode 2: $r = 0.229$, FDR-p=0.010; documentary: $r = 0.225$, FDR-p=0.005).

## Neural state dynamics are modulated by narrative event boundaries

Latent state dynamics are synchronized across individuals watching television episodes and listening to stories, which suggests that latent neural states are associated with shared cognitive states elicited by an external stimulus. How are these neural state dynamics modulated by stimulus-driven changes in cognition?

Our comedy sitcom episodes had unique event structures. Scenes alternated between two distinct storylines (A and B) that took place in different places with different characters. Each episode included 13 events (seven events of story A and six events of B) ordered in an ABAB sequence. This interleaved event structure required participants to switch between the two storylines at event boundaries and integrate them in memory to form a coherent narrative representation (*Clewett et al., 2019*; *DuBrow and Davachi, 2013*; *Zacks, 2020*).

We asked if any latent state consistently occurred at narrative event boundaries (*Figure 4A*). In both sitcom episodes, the DMN state was more likely to occur than would be expected by chance after event boundaries (~50% probability, FDR-p<0.01), complementing past work that showed the involvement of the DMN at event boundaries (*Baldassano et al., 2017*; *Chen et al., 2017*; *Reagh et al., 2020*). The base state, on the other hand, was less likely to occur after event boundaries (~10%

probability). DAN and SM state occurrences were not modulated by event boundaries (*Figure 4—figure supplement 1*). These results replicated when the SONG-defined HMM was applied to a 50 min story-listening dataset (*Chang et al., 2021b*) in which 45 events were interleaved in an ABAB sequence (*Figure 4*). A transient increase in hippocampal BOLD activity occurred after event boundaries (*Figure 4—figure supplement 1*), replicating previous work (*Baldassano et al., 2017*; *Ben-Yakov and Dudai, 2011*; *Ben-Yakov and Henson, 2018*; *Reagh et al., 2020*). Together, our results suggest that event boundaries affect neural activity not only at a regional level, but also at a whole-brain systems level.

How does brain activity transition to the DMN state at event boundaries? To investigate how event boundaries perturb neural dynamics, we compared transitions to the DMN state that occurred at event boundaries (i.e., between 5 and 15 s after boundaries) to those that occurred at the rest of the moments (non-event boundaries) (*Figure 4—figure supplement 2*). At non-event boundaries, the DMN state was most likely to transition from the base state, accounting for more than 50% of the transitions to the DMN state. Interestingly, however, at event boundaries, base-to-DMN state transitions significantly dropped while DAN-to-DMN and SM-to-DMN state transitions increased (*Figure 4C*). A repeated-measures ANOVA showed a significant interaction between the latent states and the event boundary conditions (sitcom episode 1: $F_{(2,50)} = 10.398$; episode 2: $F_{(2,52)} = 12.794$; Chang et al.: $F_{(2,48)} = 31.194$; all p-values<0.001). Thus, although the base state typically acts as a transitional hub (*Figure 2B*), neural state transitions at event boundaries are more likely to occur directly from the DAN or SM state to the DMN state without passing through the base state due to the DMN state's functional role at event boundaries. These results illustrate one way in which neural systems adaptively reconfigure in response to environmental demands.

## Neural state dynamics reflect attention dynamics in task and naturalistic contexts

In addition to changes in cognitive states, sustained attention fluctuates constantly over time (*deBettencourt et al., 2018*; *Esterman et al., 2013*; *Esterman and Rothlein, 2019*; *Fortenbaugh et al., 2018*; *Robertson et al., 1997*; *Rosenberg et al., 2020*). Previous studies showed that large-scale neural dynamics that evolve over tens of seconds capture meaningful variance in arousal (*Raut et al., 2021*; *Zhang et al., 2023*) and attentional states (*Rosenberg et al., 2020*; *Yamashita et al., 2021*). We asked whether latent neural state dynamics reflect ongoing changes in attention in both task and naturalistic contexts. To infer participants' attentional fluctuations during the gradCPT, we recorded response times (RT) to every frequent-category trial (~1 s). The RT variability time course was used as a proxy for fluctuating attentional state, with moments of less variable RTs (i.e., stable performance) indicating attentive states (*Figure 5A and B*). Paying attention to a comedy sitcom, on the other hand, involves less cognitive effort than attending to controlled psychological tasks, more akin to a 'flow'-like state compared to controlled tasks that require top-down exertion of control (*Bellana et al., 2022*; *Busselle and Bilandzic, 2009*; *Csikszentmihalyi and Nakamura, 2010*; *Kahneman, 1973*). Attending to a narrative is further affected by a rich set of cognitive processes such as emotion (*Chang et al., 2021a*; *Smirnov et al., 2019*), social cognition (*Nguyen et al., 2019*; *Yeshurun et al., 2021*), or causal reasoning (*Lee and Chen, 2022*; *Song et al., 2021b*). To assess participants' fluctuating levels of attentional engagement during the sitcom episodes and documentary, we asked participants to continuously self-report their levels of engagement on a scale of 1 (not engaging at all) to 9 (completely engaging) as they rewatched the stimuli outside the fMRI (*Figure 5A and B*; *Song et al., 2021a*).

We asked whether neural state occurrence reflected participants' attentional states. For each participant, we averaged time-resolved measures of attention based on the latent neural states that occurred at particular moments of time.

## Distinct states correspond to engaged attention during tasks and movies

Different brain states accompanied successful task performance and engaged movie watching. During the gradCPT, participants were in a high attentional state when the DMN state occurred (*Figure 5C*). Results replicated when the SONG-trained HMM was applied to the gradCPT data collected by *Rosenberg et al., 2016* (*Figure 5D*). This finding conceptually replicates previous work that showed the DMN involvement during in-the-zone moments of the gradCPT (*Esterman et al., 2013*; *Kucyi*

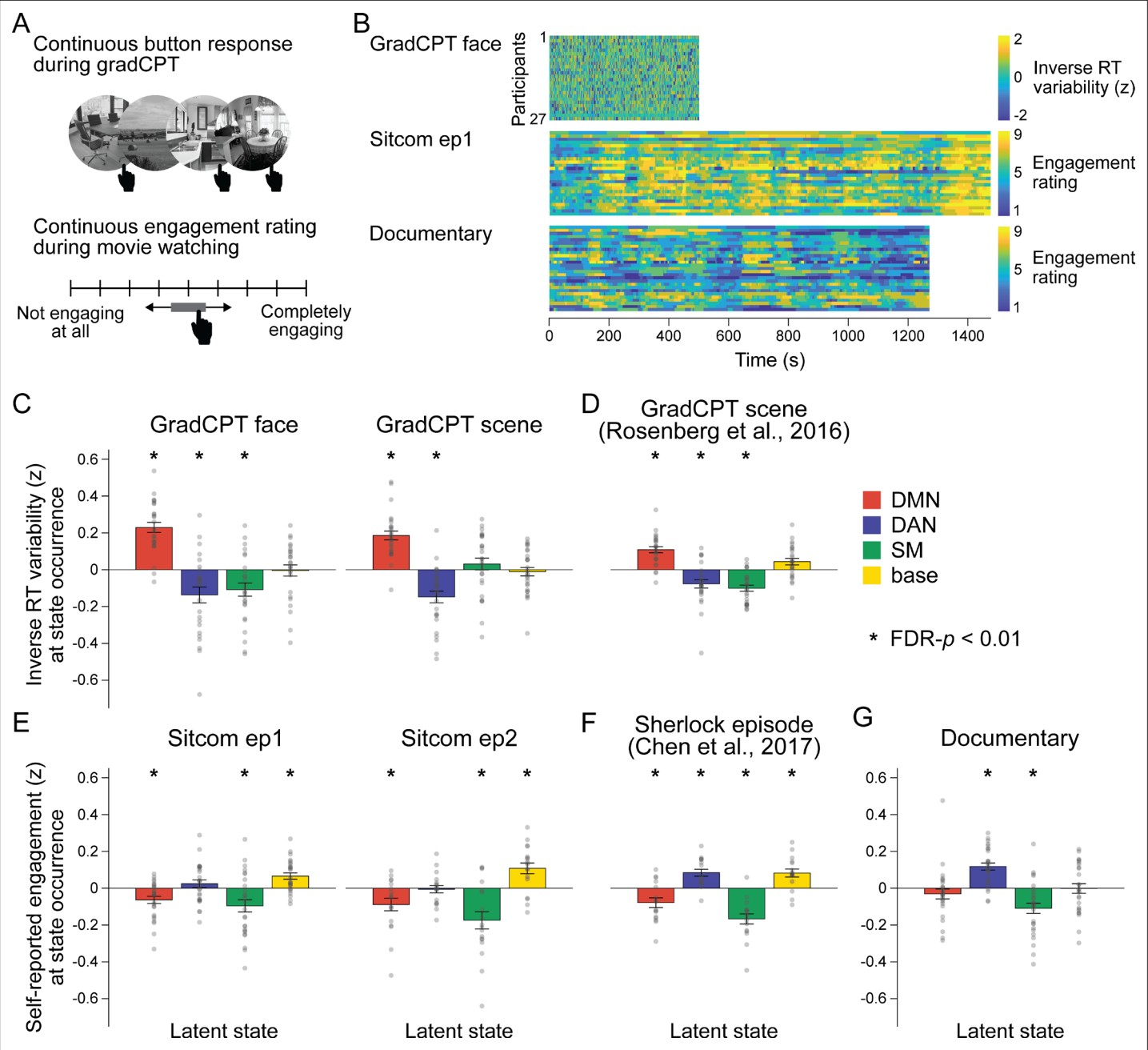

**Figure 5.** Relationship between latent neural states and attentional engagement. (**A**) Schematic illustration of the gradCPT and continuous narrative engagement rating. (Top) Participants were instructed to press a button at every second when a frequent-category image of a face or scene appeared (e.g., indoor scene), but to inhibit responding when an infrequent-category image appeared (e.g., outdoor scene). Stimuli gradually transitioned from one to the next. (Bottom) Participants rewatched the sitcom episodes and documentary after the fMRI scans. They were instructed to continuously adjust the scale bar to indicate their level of engagement as the audiovisual stimuli progressed. (**B**) Behavioral measures of attention in three fMRI conditions. Inverse RT variability was used as a measure of participants' attention fluctuation during gradCPT. Continuous ratings of subjective engagement were used as measures of attention fluctuation during sitcom episodes and documentary watching. Both measures were z-normalized across time during the analysis. (**C–G**) Degrees of attentional engagement at moments of latent state occurrence. The attention measure at every time point was categorized into which latent state occurred at the corresponding moment and averaged per neural state. The bar graphs indicate the mean of these values across participants. Gray dots indicate individual data points (participants). The mean values were compared with the null distributions in which the latent state dynamics were circular-shifted (asterisks indicate FDR-p<0.01). (**C, E, G**) Results of the fMRI runs in the SONG dataset. (**D, F**) The hidden Markov model (HMM) trained on the SONG dataset was applied to decode the latent state dynamics of (**D**) the gradCPT data by ***Rosenberg et al., 2016*** (N = 25), and (**F**) the Sherlock television watching data by ***Chen et al., 2017*** (N = 16).

The online version of this article includes the following figure supplement(s) for figure 5:

*Figure 5 continued on next page*

*Figure 5 continued*

**Figure supplement 1.** Latent state dynamics during cognitive task blocks, decoded from the hidden Markov model (HMM) trained on the SONG dataset.

_____

*et al., 2020*) and supports the role of the DMN in automated processing of both the extrinsic and intrinsic information (*Kucyi et al., 2016*; *Vatansever et al., 2017*; *Yeshurun et al., 2021*).

Other neural states indicated moments of high attention during movie watching. During comedy sitcoms, the base state was associated with engaged attention (*Figure 5E*). Results replicated when the SONG-trained HMM was applied to television episode watching data collected by *Chen et al., 2017* (N = 16) (*Figure 5F*). To our knowledge, the involvement of the base state at engaging moments of movie watching has not been reported previously. During the educational documentary, on the other hand, the DAN state was associated with engaged attention (*Figure 5G*). When watching a less engaging but information-rich documentary, focusing may require goal-directed and voluntary control of attention (*Corbetta and Shulman, 2002*). Together, the results imply that different neural states indicate engaged attention in different contexts.

## A common state underlies attention lapses during tasks and movies

In contrast to moments of engaged attention, moments of attention lapses were associated with the same brain state during gradCPT performance and movie watching. The SM state occurred during moments of poor gradCPT performance in the SONG (with the exception of the gradCPT scene run which had the shortest run duration, FDR-p=0.589; *Figure 5C*) and *Rosenberg et al., 2016* datasets (*Figure 5D*). It also occurred during periods of disengaged focus on the comedy sitcoms (*Figure 5E*), the television episode of *Chen et al., 2017* (N = 16) (*Figure 5F*), and the educational documentary (*Figure 5G*). Higher head motion was observed during the SM state compared to the three other states (*Figure 2—figure supplement 4*). However, the latent states consistently predicted attention when head motion was included as a predictor in a linear model (main effect of HMM latent states, $F > 3$, p-values<0.05 for 7 fMRI runs in *Figure 5C–G*; whereas the effect of head motion was inconsistent), demonstrating that the effects were not driven by motion alone.

To further investigate the role of the SM state, we applied the trained HMM to two external datasets, one containing gradCPT runs interleaved with fixation blocks (*Rosenberg et al., 2016*), and the other containing working memory task runs interleaved with fixation blocks (*Barch et al., 2013*; *Van Essen et al., 2013*). In both the gradCPT and working memory task, the SM state occurred more frequently during intermittent rest breaks in between the task blocks, whereas the DMN, DAN, and base states occurred prominently during the task blocks (*Figure 5—figure supplement 1*). These results suggest that the SM state indicates a state of inattention or disengagement common across task contexts.

## Discussion

Our study characterizes large-scale human fMRI activity as a traversal between latent states in a low-dimensional state space. Neural states spanned predefined gradients of functional brain organization, with the state at the center functioning as a transitional hub. These gradients explained significant variance in neural dynamics, suggesting their role as a general latent manifold shared across cognitive processes. Global desynchronization marked moments of neural state transitions, with decreases in cofluctuation of the pairwise functional networks preceding state changes. The same latent states recurred across fMRI runs and independent datasets, with distinct state-traversal patterns during rest, task, and naturalistic conditions. Neural state dynamics were synchronized across participants during movie watching and temporally aligned to narrative event boundaries. Whereas different neural states were involved in attentionally engaged states in task and naturalistic contexts, a common neural state indicated inattention in both contexts. Together, our findings suggest that human cognition and attention arise from neural dynamics that traverse latent states in a shared low-dimensional gradient space.

Taking a dynamical systems approach, systems neuroscientists have theorized that hierarchically modular systems of the brain communicate and process information dynamically (*Breakspear, 2017*). This framework, which characterizes the dynamics of systems-level interactions as a trajectory within a state space, has opened a new avenue to understanding the functional brain beyond what could be

revealed from the univariate activity of local brain regions or their pairwise connections alone (*John et al., 2022*). Although a dynamical systems approach has been adopted in non-human animal studies to understand behavior during targeted tasks (*Churchland et al., 2012*; *Kato et al., 2015*; *Mante et al., 2013*; *Sohn et al., 2019*), there is still a lack of understanding of how human cognition arises from brain-wide interactions, with a particularly sparse understanding of what gives rise to naturalistic, real-world cognition.

Using fMRI data collected in rest, task, and naturalistic contexts, we identified four latent states that tile the principal gradient axes of functional brain connectome. Are these latent states—the DMN, DAN, SM, and base states—generalizable states of the human brain? When the HMM was applied to data from each condition separately, the inferred latent states differed (*Figure 1—figure supplement 2*). However, when the HMM was applied to datasets including diverse fMRI conditions like the SONG and HCP, the four states consistently reappeared, regardless of analytical choices (*Figure 1—figure supplement 1*; *Figure 1—figure supplements 5 and 6*). We propose a framework that can unify these observations and theories: large-scale neural dynamics traverse canonical latent states in a low-dimensional manifold captured by the principal gradients of functional brain organization.

This perspective is supported by previous work that has used different methods to capture recurring low-dimensional states from spontaneous fMRI activity during rest. For example, to extract time-averaged latent states, early resting-state analyses identified task-positive and task-negative networks using seed-based correlation (*Fox et al., 2005*). Dimensionality reduction algorithms such as independent component analysis (*Smith et al., 2009*) extracted latent components that explain the largest variance in fMRI time series. Other lines of work used time-resolved analyses to capture latent state dynamics. For example, variants of clustering algorithms, such as co-activation patterns (*Liu et al., 2018*; *Liu and Duyn, 2013*), k-means clustering (*Allen et al., 2014*), and HMM (*Baker et al., 2014*; *Chen et al., 2016*; *Vidaurre et al., 2018*; *Vidaurre et al., 2017*), characterized fMRI time series as recurrences of and transitions between a small number of states. Time-lag analysis was used to identify quasiperiodic spatiotemporal patterns of propagating brain activity (*Abbas et al., 2019*; *Yousefi and Keilholz, 2021*). A recent study extensively compared these different algorithms and showed that they all report qualitatively similar latent states or components when applied to fMRI data (*Bolt et al., 2022*). While these studies used different algorithms to probe data-specific brain states, this work and ours report common latent axes that follow a long-standing theory of large-scale human functional systems (*Mesulam, 1998*). Neural dynamics span principal axes that dissociate unimodal to transmodal and sensory to motor information processing systems.

Prior systems neuroscience research on low-dimensional brain states was primarily performed on data from rest or a single task. Thus, the extent to which a latent manifold underlying brain states is common or different across contexts was unknown. It was also unclear how brain states reflected cognitive dynamics. Our results show that neural dynamics in different cognitive contexts can be coarsely understood as traversals between latent states in a context-general manifold. However, the state dynamics, or most likely 'paths' between states, differ with context and functional demands, potentially giving rise to our diverse and flexible cognitive processes.

Our study adopted the assumption of low dimensionality of large-scale neural systems, which led us to intentionally identify only a small number of states underlying whole-brain dynamics. Importantly, however, we do not claim that the four states will be the optimal set of states in every dataset and participant population. Instead, latent states and patterns of state occurrence may vary as a function of individuals and tasks (*Figure 1—figure supplement 2*). Likewise, while the lowest dimensions of the manifold (i.e., the first two gradients) were largely shared across datasets tested here, we do not argue that it will always be identical. If individuals and tasks deviate significantly from what was tested here, the manifold may also differ along with changes in latent states (*Samara et al., 2023*). Brain systems operate at different dimensionalities and spatiotemporal scales (*Greene et al., 2023*), which may have different consequences for cognition. Asking how brain states and manifolds—probed at different dimensionalities and scales—flexibly reconfigure (or not) with changes in contexts and mental states is an important research question for understanding complex human cognition.

Previous studies reported functional relevance of latent state dynamics during controlled (*Cornblath et al., 2020*; *Reddy et al., 2018*; *Shine et al., 2019a*; *Taghia et al., 2018*; *Yamashita et al., 2021*) and naturalistic tasks (*Song et al., 2021b*; *van der Meer et al., 2020*). The current study aimed to unify these findings by generalizing the latent state model to multiple fMRI runs and datasets

spanning rest, task, and naturalistic contexts. Intriguingly, the latent states commonly occurred in every scan type (*Figure 3B*), but their functional roles differed depending on context. For example, during monotonous tasks that required constant exertion of sustained attention, the DMN state accompanied successful, stable performance whereas the DAN state characterized suboptimal performance (*Figure 5C and D*). The antagonistic activity and functional relationship between the DMN and DAN has been reported in past studies that used resting-state (*Buckner et al., 2008*; *Fox et al., 2005*) or task fMRI (*Esterman et al., 2013*; *Kelly et al., 2008*; *Kucyi et al., 2020*). In contrast, in naturalistic contexts, the DMN state indicated low attentional engagement to narratives (*Figure 5E and F*) and tended to follow event boundaries (*Figure 4A and B*). The DAN state, on the other hand, indicated high attentional engagement during documentary watching (*Figure 5G*) and was not modulated by event boundaries (*Figure 4—figure supplement 1*). Our results indicate that the functional relationship between the DMN and DAN states shows more nuanced dependence to contexts. (Though our observations align with previous work on the functional roles of the default mode and dorsal attention networks, it is important to keep in mind that the two states are not just characterized by activation of these networks but by patterns of activation and covariation of the whole brain networks. They should be interpreted as 'states' rather than isolated functional networks.) The findings highlight the need to probe both the controlled and naturalistic tasks with dense behavioral sampling to fully characterize the functional roles of these neural states (*Song and Rosenberg, 2021*).

In contrast to the context-specific DMN and DAN states, the SM state consistently indicated inattention or disengagement. The SM state occurred during poor task performance and low narrative engagement (*Figure 5*) as well as during intermittent task breaks (*Figure 5—figure supplement 1*). The result implies that whereas the optimal neural state may vary with information processing demands, a suboptimal state is shared across contexts.

Previous work showed that time-resolved whole-brain functional connectivity (i.e., paired interactions of more than a hundred parcels) predicts changes in attention during task performance (*Rosenberg et al., 2020*) as well as movie watching and story listening (*Song et al., 2021a*). Future work could investigate whether functional connectivity and the HMM capture the same underlying 'brain states' to bridge the results from the two literatures. Furthermore, though the current study provided evidence of neural state dynamics reflecting attention, the same neural states may, in part, reflect fluctuations in arousal (*Chang et al., 2016*; *Zhang et al., 2023*). Complementing behavioral studies that demonstrated a nonlinear relationship between attention and arousal (*Esterman and Rothlein, 2019*; *Unsworth and Robison, 2018*; *Unsworth and Robison, 2016*), future studies collecting behavioral and physiological measures of arousal can assess the extent to which attention explains neural state dynamics beyond what can be explained by arousal fluctuations.

Past resting-state fMRI studies have reported the existence of the base state. *Chen et al., 2016* used the HMM to detect a state that had 'less apparent activation or deactivation patterns in known networks compared with other states.' This state had the highest occurrence probability among the inferred latent states, was consistently detected by the model, and was most likely to transition to and from other states, all of which mirror our findings here. The authors interpret this state as an 'intermediate transient state that appears when the brain is switching between other more reproducible brain states.' The observation of the base state was not confined to studies using HMMs. *Saggar et al., 2022* used topological data analysis to represent a low-dimensional manifold of resting-state whole-brain dynamics as a graph, where each node corresponds to brain activity patterns of a cluster of time points. Topologically focal 'hub' nodes were represented uniformly by all functional networks, meaning that no characteristic activation above or below the mean was detected, similar to what we observe with the base state. The transition probability from other states to the hub state was the highest, demonstrating its role as a putative transition state.

However, the functional relevance of the base state to human cognition had not been explored previously. We propose that the base state, a transitional hub (*Figure 2B*) positioned at the center of the gradient subspace (*Figure 1D*), functions as a state of natural equilibrium. Transitioning to the DMN, DAN, or SM states reflects incursion away from natural equilibrium (*Deco et al., 2017*; *Gu et al., 2015*), as the brain enters a functionally modular state. Notably, the base state indicated high attentional engagement (*Figure 5E and F*) and exhibited the highest occurrence proportion (*Figure 3B*) as well as the longest dwell times (*Figure 3—figure supplement 1*) during naturalistic movie watching, whereas its functional involvement was comparatively minor during controlled tasks.

This significant relevance to behavior verifies that the base state cannot simply be a by-product of the model. We speculate that susceptibility to both external and internal information is maximized in the base state—allowing for roughly equal weighting of both sides so that they can be integrated to form a coherent representation of the world—at the expense of the stability of a certain functional network (*Cocchi et al., 2017*; *Fagerholm et al., 2015*). When processing rich narratives, particularly when a person is fully immersed without having to exert cognitive effort, a less modular state with high degrees of freedom to reach other states may be more likely to be involved. The role of the base state should be further investigated in future studies.

This work provides a framework for understanding large-scale human brain dynamics and their relevance to cognition and behavior. Neural dynamics can be construed as traversals across latent states along the low-dimensional gradients, driven by interactions between functional networks. The traversals occur adaptively to external and internal demands, reflecting ongoing changes of cognition and attention in humans.

## Materials and methods
### SitcOm, Nature documentary, Gradual-onset continuous performance task (SONG) neuroimaging dataset
#### Participants
Twenty-seven participants were recruited in Korea (all native Korean speakers; two left-handed, 15 females; age range 18–30 y with mean age 23 ± 3.16 y). Participants reported no history of visual, hearing, or any form of neurological impairment, passed the Ishihara 38 plates color vision deficiency test (https://www.color-blindness.com/ishihara-38-plates-cvd-test) for red-green color blindness, provided informed consent before taking part in the study, and were monetarily compensated. The study was approved by the Institutional Review Board of Sungkyunkwan University. None of the participants were excluded from analysis.

#### Study overview
Participants visited twice for a 3 hr experimental session per day. Sessions were separated by approximately 1 wk on average (mean 8.59 ± 3.24 d, range 2–15 d). Two participants returned for an additional scan and behavioral session because technical difficulties prevented them from completing the experiment within the 2 d.

During the first scan session, participants watched the first episode of a sitcom as well as a documentary clip during fMRI. Scan order was counterbalanced. One participant's sitcom episode 1 fMRI run was not analyzed because the data were not saved. Structural T1 images were collected after EPI acquisitions. Immediately after the MRI scan session, participants completed behavioral tasks in a different room. They first completed free recall of the two movie clips in the order of viewing. These data are not analyzed here but were used to confirm that the participants were awake during the scans. Participants then were asked to complete continuous engagement ratings while rewatching the same audiovisual stimuli in the same order. This fMRI experiment lasted approximately 1 hr, and the post-scan behavioral experiment lasted approximately 1.5–2 hr.

The second scan session began with two 10 min resting state runs in which the participants were asked to fixate on a centrally presented black cross on a gray background. Next, participants completed the gradCPT with face images, watched the second sitcom episode, and performed the gradCPT with scene images. After fMRI, participants completed two runs of a recognition memory task for the scene images that were viewed during gradCPT and a free recall of the sitcom episode 2. These data were not analyzed here. The continuous engagement rating for the second sitcom episode was not collected during this session due to time limitations. 18 of the 27 participants returned to the lab to complete the continuous engagement rating task for the second sitcom episode. One participant's behavioral data during the gradCPT run with face images were not saved. The fMRI experiment lasted approximately 1.5 hr, and the post-scan behavioral experiment lasted approximately 1 hr.

## Sitcom episodes and documentary watching

### Stimuli

Two episodes of the comedy sitcom and one educational documentary clip were used as audiovisual stimuli. The sitcom, *High Kick Through the Roof*, is a South Korean comedy sitcom that was aired in 2009–2010 on a public television channel, MBC (https://en.wikipedia.org/wiki/High_Kick_Through_the_Roof). The duration of the first episode was 24 min 36 s, and the second episode was 24 min 15 s. These episodes were chosen because the narrative followed an interleaved ABAB sequence. Events of a story A (e.g., which took place in a forest and centered around two sisters) happened independently from events of a story B (e.g., which took place in a city and followed members of a large family), and the two storylines occurred in different times and places and included different characters. To avoid a transient increase in fMRI activity upon a sudden video presentation, we included a 30 s of a dummy video clip from the Minions Mini Movies (2019) (https://www.youtube.com/watch?v=sL3kLxsy-Lg) prior to the presentations of the sitcom episodes that was discarded in analysis.

*Rivers of Korea, Part 1* (21 min 33 s) is a documentary that aired in 2020 by a public educational channel EBS Docuprime in South Korea. The documentary introduces the history and geography of the two largest Korean rivers, the Han and Nakdong Rivers. This stimulus was chosen because while it has rich and dynamically changing audiovisual and narrative content, it elicits an overall low degree of engagement due to its educational purpose (although individuals may vary in the degree to which they find it engaging). The first 22 s of the documentary was not included in the analysis to account for a sudden increase in brain activity upon video presentation.

The audiovisual stimuli were presented at a visual dimension of 1280 × 720 and frame rate of 29.97 Hz on a black background. 30 s of center fixation was included at the end of every naturalistic stimulus run. No additional task was given to participants during fMRI except for an instruction to stay vigilant and attentive to the video.

### Continuous engagement rating

Participants rewatched the videos in a behavioral testing room while they were instructed to continuously adjust the scale bar from scale of 1 (not at all engaging) to 9 (completely engaging) that was visible on the bottom of the monitor. Participants were instructed to report their experience as closest to when they have watched the stimulus during the fMRI. The definition of engagement was given to participants following *Song et al., 2021a* as: I find the story engaging when (i) I am curious and excited to know what's coming up next, (ii) I am immersed in the story, (iii) My attention is focused on the story, and (iv) The events are interesting; whereas I find the story not engaging when (i) I am bored; (ii) Other things pop into my mind, like my daily concerns or personal events; (iii) My attention is wandering away from the story; (iv) I can feel myself dozing off; and (v) The events are not interesting. Participants were encouraged to adjust the scale bar whenever their subjective engagement changed during the sitcom episodes or documentary. All participants completed a practice session with a clip from a Korean YouTube channel (https://www.youtube.com/c/VIVOTVchannel). Stimuli were presented with Psychopy3 (*Peirce, 2007*) on a MacBook 13-inch laptop. Participants were given freedom to turn on or turn off the light or have the headphone or speaker on. Upon completion of continuous engagement ratings, participants were asked to give an overall engagement score of the stimulus using the same 1–9 Likert scale.

Continuous engagement rating time courses, ranging from 1 to 9, were *z*-normalized across time per participant and convolved with the canonical hemodynamic response function to be related with the neural state dynamics.

## Gradual-onset continuous performance task (gradCPT)

### Task with face images

Grayscale face images (nine females and one male unique faces) were selected from the MIT Face Database (*Rosenberg et al., 2013*; *Russell, 2009*), cropped to a circle at a visual dimension of 300 × 300, and presented on a gray background. 500 trials (450 female and 50 male face trials) were included in the run, with each unique face image appearing 50 times in a random sequence. No repeats of the images were allowed on consecutive trials. On each trial, an image gradually transitioned from one to the next using a linear pixel-by-pixel interpolation. The transition took 800 ms, and the intact face

image stayed for 200 ms when fully cohered. The task was to press a button on each trial when a female face appeared (90% of trials) but to inhibit making a response when a male face appeared (10% of trials). A fixation cross appeared for the first 1 s of the run, and the trial sequence started with a dummy stimulus (scrambled face). The run ended with a 10 s of center-fixation and lasted 8 min 33 s in total. A practice session was completed with the same face images prior to the scans.

### Task with scene images

Colored scene images (300 indoor, 300 outdoor) were selected from the SUN database (**deBettencourt et al., 2018**; **Xiao et al., 2010**). The stimulus was presented at a visual dimension of 500 × 500 on a gray background. Each individual saw 360 trial-unique images in a random order, with 300 (83.33%) coming from a frequent category (e.g., indoor) and 60 (16.67%) from an infrequent category (e.g., outdoor). Whether the indoor or outdoor scenes corresponded to the frequent category was counterbalanced across participants. Images transitioned in a pixelwise interpolation, with a transition occurring over 500 ms and the intact image lasting 700 ms. Participants were asked to press '1' for frequent-category and '2' for infrequent-category scene images. A fixation cross appeared at first 1 s of the run, and the trial started with a dummy stimulus (scrambled scene). The run ended with a 10 s of center-fixation and lasted 7 min 25 s in total. A practice session was completed with a different set of scene images prior to the scans.

### Response time assignment algorithm

Each gradCPT trial included moments when an image was interpolated with the previous trial's image followed by the fully cohered image. A maximum of two responses were recorded per trial. For most trials, a single response or no response was recorded within the trial time window. However, if two responses were recorded in a trial (2.35% and 0.27% of all trials from tasks with face and scene images) and the response for the previous trial was missing, then the first response was regarded as a response for the previous trial and the second response as the response for the current trial. In cases when two responses were recorded but the response for the previous trial was not missing, or when the response for a single response trial happened before 40% of image coherence, we chose a response that favored a correct response. For trials where there were errors in image presentation (two participants during gradCPT scene runs: 1 trial and 18 consecutive trials out of 360 trials) or participants did not respond for consecutive trials (one participant during gradCPT face run: 66 trials out of 500 trials), the accuracy and response times for that trials were treated as NaNs.

### Response time variability

To calculate an RT variability time course for each gradCPT run, the response times for incorrect and no-response trials were treated as NaNs, which were then filled by 1D linear interpolation. The response time course was linearly detrended, and RT variability was calculated by taking the deviance from the mean RT at every TR. Because the trial duration of the gradCPT scene runs was 1.2 s, the RT variability time course was resampled to match the TR resolution of 1 s. Each participant's mean RT variability was appended as the value corresponding to the first TR. The RT variability time course was z-normalized across time within run and convolved with the canonical hemodynamic response function to be related with the neural state dynamics.

## FMRI image acquisition and preprocessing

Participants were scanned with a 3T scanner (Magnetom Prisma; Siemens Healthineers, Erlangen, Germany) with a 64-channel head coil. Anatomical images were acquired using a T1-weighted magnetization-prepared rapid gradient echo pulse sequence (repetition time [TR] = 2200 ms, echo time [TE] = 2.44 ms, field of view = 256 mm × 256 mm, and 1 mm isotropic voxels). Functional images were acquired using a T2*-weighted echo planar imaging (EPI) sequence (TR = 1000 ms, TE = 30 ms, multiband factor = 3, field of view = 240 mm × 240 mm, and 3 mm isotropic voxels, with 48 slices covering the whole brain). The number of TRs per run are as follows: resting-state run 1 (602 TR) and run 2 (602 TR), gradCPT with face (513 TR) and scene images (445 TR), sitcom episode 1 (1516 TR), episode 2 (1495 TR), and documentary (1303 TR). Visual stimuli were projected from a Propixx projector (vPixx Technologies, Bruno, Canada), with a resolution of 1920 × 1080 pixels and a refresh

rate of 60 Hz. Auditory stimuli were delivered by MRI compatible in-ear headphones (MR Confon; Cambridge Research Systems, Rochester, UK).

Structural images were bias-field corrected and spatially normalized to the Montreal Neurological Institute (MNI) space using FSL. The first two images of the resting-state and gradCPT runs, 30 for sitcom episodes and 22 for documentary were discarded to allow the MR signal to achieve T1 equilibration. Functional images were motion-corrected using the six rigid-body transformation parameters. The functional images were intensity-normalized, and the FMRIB's ICA-based X-noiseifier (FIX) was applied to automatically identify and remove noise components (*Griffanti et al., 2017*; *Griffanti et al., 2014*; *Salimi-Khorshidi et al., 2014*). The images were registered to MNI-aligned T1-weighted images. We additionally regressed out low-frequency components (high-pass filtering, $f > 0.009$ Hz), linear drift, and the global signal. The raw datasets of the SONG, *Rosenberg et al., 2016*, and *Chen et al., 2017* were all preprocessed with the same pipeline. Results replicated when processing included band-pass filtering in place of high-pass filtering ($0.009 < f < 0.08$ Hz) and when preprocessing did not include global signal regression (*Figure 1—figure supplement 5*). All analyses were conducted in volumetric space.

## Human Connectome Project (HCP) dataset

We used 3-Tesla and 7-Tesla data from 184 young adult participants in the HCP dataset. 3-Tesla data included four resting-state runs (*REST1* and *REST2* in the left-to-right and right-to-left phase encoding directions) and two runs each of the seven tasks (*EMOTION*, *GAMBLING*, *LANGUAGE*, *MOTOR*, *RELATIONAL*, *SOCIAL*, and *WORKING MEMORY*). 7-Tesla data included four resting-state runs (*REST1_PA*, *REST2_AP*, *REST3_PA*, *REST4_AP*) and four movie-watching runs (MOVIE1_*AP*, MOVIE2_*PA*, MOVIE3_*PA*, MOVIE4_*AP*). The combined data included 8400 TRs of the resting-state runs, 3880 TRs of task runs, and 3655 TRs of the movie-watching runs per participant. Of these 184 individuals, we excluded 19 who had not completed any of the scan runs, and 6 whose scan run was aborted earlier than others. Additionally, 40 participants' data were discarded due to excessive head motion; having at least one fMRI run with more than 20% of the time points' framewise displacement (FD) ≥ 0.5 or mean FD ≥ 0.5. This resulted in the analysis of 119 participants in total. We downloaded the MNI-aligned, minimally preprocessed structural and functional MRI images from the HCP repository (*Glasser et al., 2013*). Additionally, the global signal, white matter, and cerebrospinal fluid time courses, 12 head motion parameters (provided by Movement_Regressors.txt), and a low-frequency component (high-pass filtering, $f > 0.009$ Hz) were regressed from the data. For details on fMRI image acquisitions and task procedures, see *Barch et al., 2013*; *Finn and Bandettini, 2021*; *Van Essen et al., 2013*.

## Hidden Markov model (HMM)

We used the HMM to characterize the dynamics of latent neural states that underlie large-scale functional brain activity (*Rabiner and Juang, 1986*). First, we parcellated the whole brain into 17 cortical networks (*Yeo et al., 2011*) and 8 subcortical regions (*Tian et al., 2020*) and averaged the BOLD time series of the voxels that corresponded to these parcels. The 25 parcel time courses of all participants' every run were z-normalized within-run and concatenated.

Expectation-maximization (*Dempster et al., 1977*) of the forward-backward algorithm was used to estimate the optimal model parameters: (i) the emission probability $p(y_t \mid x_t)$ of the observed fMRI time series { $y_1 \dots y_T$ } from the hidden latent sequence { $x_1 \dots x_T$ }, and (ii) the first-order Markovian transition probabilities $p(x_t = s_i \mid x_{t-1} = s_j)$ for $1 \leq i,\ j \leq K$. The emission probability was modeled using a mixture Gaussian density function (hmmlearn.hmm.GaussianHMM), such that $p(y_t \mid x_t = s)$ $\sim \mathfrak{N}\left(x \mid \mu_s,\ \sigma_s^2\right) = \frac{1}{\sqrt{2\pi\sigma_s^2}} e^{-\frac{(x-\mu_s)^2}{2\sigma_s^2}}$ per discrete $K$ number of latent state $s \in \{\ s_1 \dots s_K\ \}$, where $\mu_s = \frac{1}{N}\sum_{i=1}^{N} y_i$ and $\sigma_s^2 = \frac{1}{N}\sum_{i=1}^{N}\left(y_i - \mu_s\right)^2$ for a set of observed fMRI time steps { $y_1 \dots y_N$ } identified as the latent state $s$. The $\mu_s$ and $\sigma_s^2$ represent the mean activation and covariance patterns of the 25 parcels per each state (*Figure 1A*). The transitions between the hidden states were assumed to have a form of a first-order Markov chain, such that if $a_{ij}$ represents the probability of transitioning from state $i$ to state $j$, then $\sum_{i=1}^{K} a_{ij} = 1$. The inference procedure terminated if there was no longer a gain in log-likelihood during the re-estimation process of the forward-backward algorithm or if the number

of maximum 1000 iterations was reached. We initialized the HMM parameters using the output of k-means clustering to overcome the problem of falling into a local minima (sklearn.cluster.kMeans).

The estimated transition and emission probabilities were applied to decode the most probable latent state sequence conditioned on the observed fMRI time series using a Viterbi algorithm (*Rezek and Roberts, 2005*). The Viterbi algorithm estimates the probability of each latent state being the most likely state at a specific time point. We chose the state with the highest probability at every time step (TR), thus discretizing the latent sequence.

To choose the optimal number of latent states (K), a hyperparameter that needed to be selected in advance, we conducted the HMM in a leave-one-subject-out cross-validated manner where we trained the HMM on all participants but one to infer transition and emission probabilities, and applied the HMM to decode the latent state dynamics of the held-out participant. A Calinski–Harabasz score was compared across the choice of K from 2 to 10 (*Calinski and Harabasz, 1974*; *Gao et al., 2021*; *Song et al., 2021b*; *Figure 1—figure supplement 1*). The K with the largest mean Calinski–Harabasz score across cross-validations was selected, and we conducted the HMM on all participants' data with the chosen K. The HMM inference and decoding procedure was repeated 10 times, and the instance with the maximum expected likelihood was chosen as a final result.

The surrogate latent sequence was generated by having 25-parcel time series circular-shifted across time respectively for each parcel, thereby disrupting meaningful covariance between parcels while retaining temporal characteristics of the time series, and applying the same HMM fitting and decoding algorithms 1000 times. The maximum number of estimations was set as the number of iterations that was reached during the actual HMM procedure (1000 for SONG and 248 for HCP).

Unless otherwise noted, the HMM parameter inference was conducted on the SONG and HCP datasets respectively and the model decoded latent state sequence of the same dataset. However, for analyses validating the functional roles of the latent states, the parameters inferred from SONG data were used to decode the latent state sequence of external datasets collected by (*Figure 4B*), *Rosenberg et al., 2016* (*Figure 5D*, *Figure 5—figure supplement 1A and B*), *Chen et al., 2017* (*Figure 5F*), and working memory runs of the HCP dataset (*Barch et al., 2013*; *Van Essen et al., 2013*; *Figure 5—figure supplement 1C and D*).

*Figure 1B* shows mean activity and covariance patterns derived from the Gaussian emission probability estimation. The brain surfaces were visualized with nilearn.plotting.plot_surf_stat_map. The parcel boundaries in *Figure 1B* are smoothed from the volume-to-surface reconstruction.

Covariance strength was operationalized as the sum of the absolute covariance weights of all possible pairwise edges. Covariance strength calculated for each latent state was compared to chance distributions generated from covariance matrices estimated from the HMMs conducted on the circular-shifted 25 parcel time series (1000 iterations, two-tailed non-parametric permutation tests, FDR-corrected for the number of latent states).

## Predefined and data-driven functional connectivity gradients

The gradients of the cortical and subcortical voxels estimated by *Margulies et al., 2016* were downloaded from the repository (https://identifiers.org/neurovault.collection:1598). The gradient values of the voxels within each of the 25 parcels were averaged to represent each parcel's position in the gradient space. To situate the latent states in Margulies et al.'s (2016) gradient axes, we took the mean of element-wise product between these gradient values and the mean activity loadings of the 25 parcels inferred by the HMM. To ask whether the four latent states are maximally distant from one another, we computed the Euclidean distance between every pair of latent states in the two-dimensional gradient space. The mean distance between all pairs of states was compared to a chance distribution where the null latent states were derived from HMM inferences on the circular-shifted 25 parcel time series (1000 iterations, two-tailed non-parametric permutation tests). Furthermore, to test whether the latent states were positioned at more extreme ends of the gradients than expected by chance, we situated the same null latent states in the gradient space. Each latent state's position on each gradient axis was compared to a chance distribution. The significance was FDR-corrected for eight comparisons (four latent states on two-dimensional axes).

We evaluated how well these predefined gradients captured large-scale neural dynamics compared to data-driven gradients defined from the SONG and HCP datasets. To do so, we extracted fMRI time series from the 1000 cortical ROIs of *Schaefer et al., 2018* and 54 subcortical ROIs of *Tian et al.,*

*2020*. The 1054 ROI time series were *z*-normalized across time, and the time series from multiple scan runs were concatenated within each participant. All participants' 1054 ROI-by-ROI functional connectivity matrices were averaged. As in *Margulies et al., 2016*, the average functional connectivity matrix was thresholded row-wise at 90% for sparseness. The affinity matrix was computed using cosine distance and then decomposed using diffusion embedding. The variance of the functional connectome explained by each gradient was computed by taking the ratio of its eigenvalue to the sum of all eigenvalues.

First, to compare the predefined gradients and data-specific gradients, we calculated Pearson's correlations across the two 1054 parcels' gradient embeddings. Next, we computed the variance that the predefined and data-specific gradients explained in the 1054 ROI time series of the SONG and HCP datasets. The fMRI time series were projected onto the first two gradient axes by taking the mean of element-wise product between the fMRI time series (time × 1054 ROIs) and the gradient embeddings (1054 ROIs × 2). The explained variance was calculated by the mean of squared Pearson's correlations ($r^2$) between the 1054 ROI fMRI time series (time × 1054 ROIs) and the gradient-projected time series (time × 2). All participants' all runs were concatenated to compute one $r^2$ per gradient axis. The explained variance of the first two gradients was equal to the sum of two $r^2$ values.

## Cofluctuation time course time-aligned to neural state transitions

Cofluctuation, the absolute element-wise product of the normalized activity in two regions, was computed based on *Faskowitz et al., 2020*; *Zamani Esfahlani et al., 2020*. We normalized the time courses of the two regions among the 25 parcels, $x_i$ and $x_j$ ($i$=1 ... T and $j$ = 1 ... T), such that $z_i = \frac{x_i - \mu_i}{\sigma_i}$, where $\mu_i = \frac{1}{T} \sum_t x_i(t)$ and $\sigma_i = \sqrt{\frac{1}{T-1} \sum_t \left(x_i(t) - \mu_i\right)^2}$. The cofluctuation time series between $z_i$ and $z_j$ was computed as the absolute of the element-wise product, $| z_i \cdot z_j |$, which represents the magnitude of moment-to-moment cofluctuations between region $i$ and $j$ based on their baseline activities. Cofluctuation was computed for every pair of parcels, resulting in a time-resolved, 25 (parcel) × 25 × T (number of TRs) matrix for each fMRI run, with a symmetric matrix at every time point.

We categorized the 25-parcel pairs to cortico-cortical (136 pairs), cortico-subcortical (136 pairs), and subcortico-subcortical (28 pairs) connection categories. The cofluctuation time courses were time-aligned to multiple moments of state transitions indicated by the HMM inference, which were averaged within a participant. The time-aligned mean cofluctuation of each pair was then averaged across all runs of the entire participants.

The chance distributions were created in two ways. First, cofluctuation time courses were time-aligned to the circular-shifted indices of neural state transitions (1000 iterations; *Figure 2A*). Second, we circular-shifted the 25-parcel time series, thereby disrupting their covariance structure, and conducted HMM inference on these null time series (1000 iterations; *Figure 2—figure supplement 2*). The time-aligned cofluctuation of every pair of parcels was averaged per category and was compared to chance distributions using *z*-statistics and two-tailed non-parametric permutation tests. The significance at each time point was FDR-corrected for the number of time points (i.e., –3 to 3 from the onset of new latent states). Furthermore, to compare the degrees of cofluctuation change between cortico-cortical, cortico-subcortical, and subcortico-subcortical connection categories, mean cofluctuation difference at time *t-1* and *t+3* was taken per category and the values were compared across categories using paired Wilcoxon signed-rank tests (FDR-corrected for three pairwise comparisons).

## Neural state transition probabilities

The *T-1* number of transitions in each participant's latent state sequence were categorized based on which state it transitioned from (at *t-1*) and which state it transitioned to (at *t*) as a 4 ('from' state) × 4 ('to' state) transition-count matrix. We controlled for the number of state occurrences by either dividing each element by the sum of each row, which identified the probabilities of transitioning 'to' one of the four latent states (*Figure 2B*), or dividing by the sum of each column, which identified the probabilities of transitioning 'from' one of the four latent states (*Figure 2—figure supplement 3*). The transition probability matrices estimated from every participant's every run were averaged. The chance distribution was created by conducting the HMM fits on the circular-shifted 25 parcel time series (1000 iterations). Significance was computed for each state pair of the transition matrix using the two-tailed non-parametric permutation tests (FDR-corrected for 16 pairs).

## Global cofluctuation of the latent states

Cofluctuation time courses of all pairs of parcels were averaged within a run of each participant. This global cofluctuation measure at every TR was categorized based on the HMM latent state identification, which was then averaged per state (*Figure 2C*). Cofluctuation values of the base state of all participants' entire runs were compared with the DMN, DAN, and SM states' using the paired *t*-tests (FDR-corrected for three comparisons). The values at each latent state were averaged and compared to a chance distribution in which the analysis was repeated with a circular-shifted latent state sequence (1000 iterations, two-tailed non-parametric permutation tests, FDR-corrected for the number of states).

## Fractional occupancy of the latent states

Using the HMM-derived latent state sequence, we calculated fractional occupancy of the latent states for all participants' every run. Fractional occupancy is the probability of latent state occurrence over the entire fMRI scan sequence, with a chance value of 25% when K = 4. The mean of all participants' fractional occupancy values was bootstrapped (10,000 iterations) for visualization in *Figure 3B*.

## Pairwise participant similarity of the latent state sequence

The similarity between pairs of participants' latent state sequences was computed as the ratio of the times when the same state occurred over the entire time course. The mean similarity was compared to the chance distribution in which participants' neural state dynamics were circular-shifted 1000 times (FDR-corrected for the number of fMRI runs). To compare the degree of synchrony across conditions, we bootstrapped the same pairs of participants with replacement ($_NC_2$ iterations, where N = number of participants) in paired conditions and took the differences of bootstrapped participant pairs' latent state sequence similarities. The median of these differences was extracted 1000 times, and the distribution was compared to 0 non-parametrically.

## Neural state dynamics at narrative event boundaries

Narrative event boundaries were marked by the experimenter at moments in the sitcom episodes when an event of a storyline transitioned to another event of a different story. Both sitcom episodes comprised 13 events (seven events of story A and six events of story B), and thus 12 event boundaries. The latent state sequences at *t-2* and *t+20* TRs from each of the 12 event boundaries were extracted, and the mean probability of state occurrence across these event boundaries was computed for every latent state within a participant. The probabilities were then averaged across participants. The state occurrence probability at every time step was compared to a chance distribution that was created by relating neural state dynamics to circular-shifted moments of event boundaries (1000 times, two-tailed non-parametric permutation tests, FDR-corrected for number of time points). An audio-story listening data of *Chang et al., 2021b* comprised 45 interleaved events. Because TR resolution (TR = 1.5 s) was different from the SONG dataset, latent neural states from *t-2* to *t+16* TRs from event boundaries were used in analysis.

Next, we compared transitions made to the DMN state at event boundaries to transitions made to the DMN state at moments other than event boundaries. We categorized every transition to the DMN state (i.e., from the DAN, SM, or base state) based on whether it occurred 5–15 TRs (for *Chang et al., 2021b*, 4–12 TRs) after a narrative event boundary or not. The proportions of DAN-to-DMN, SM-to-DMN, and base-to-DMN state transitions at event boundaries and non-event boundaries were compared using paired Wilcoxon signed-rank tests (FDR-corrected for three comparisons). The interaction between the DMN-preceding latent states and event boundary conditions was tested using the repeated-measures ANOVA.

## Neural state dynamics related to attention dynamics

We measured participants' attention fluctuations during gradCPT and movie-watching fMRI scans. Fluctuations during gradCPT performance were inferred from the inverted RT variability time course (*z*-normalized) collected concurrently during the gradCPT scans with face and scene images, as well as during gradCPT in the *Rosenberg et al., 2016* dataset. Continuous engagement rating time courses (*z*-normalized) collected after the sitcom episode and documentary watching scans were used to infer changes in the degree to which the naturalistic stimuli were engaging over time. Engagement ratings

of the Sherlock dataset (*Chen et al., 2017*) were collected by *Song et al., 2021a* from an independent group of participants.

To relate attentional dynamics to the occurrence of neural states, we categorized each person's attention measure at every TR based on the HMM latent state identification, which was averaged per state. The mean attention measures of the four states were averaged across participants and compared to a chance distribution in which the attention measures were circular-shifted to be related to the latent state dynamics (1000 iterations, two-tailed non-parametric permutation tests, FDR-corrected for the number of states). For Sherlock dataset only, a single group-average engagement time course was related to the latent state sequence of each fMRI participant because we did not have fMRI participant-specific behavioral ratings.

Linear mixed-effects models were conducted on the seven fMRI runs (*Figure 5C–G*), where the model predicted attention measure at every time step from the inferred HMM state indices and head motion (framewise displacement computed after fMRI preprocessing). The participant index was treated as a random effect. The significance of the two main effects and their interaction were computed using ANOVA.

To investigate the role of the SM state, we analyzed the gradCPT data collected by *Rosenberg et al., 2016* and two sessions of the HCP working memory task (*Figure 5—figure supplement 1*). The *Rosenberg et al., 2016* dataset (N = 25) included gradCPT task blocks separated by intervening fixation blocks. The working memory task run of the HCP dataset included 2-back and 0-back working memory task blocks and fixation blocks (N = 119). Nine participants' left-to-right phase encoding runs and 11 participants' right-to-left phase encoding runs in the HCP working memory dataset were discarded in the analysis either because their block orders differed from the other HCP participants' or because the experiment log was not saved in the dataset repository. Latent state fractional occupancy values were computed for each task block within a participant. Comparisons of a state's fractional occupancies across block types were based on paired *t*-tests (FDR-corrected for the number of states). The interaction between latent neural states and task block types was tested using repeated-measures ANOVA.

## Acknowledgements

We thank Kyung Soo Song for suggestions on audiovisual narrative stimuli. We thank JeongJun Park for help with fMRI and behavioral data collection, Jeongwon Shin for help with behavioral data collection, Boohee Choi for help with fMRI data collection, and Wooyoung Cho and Sunghyun Ban for their help with recall transcriptions. We thank Kyeong-Jin Tark and members of the Awh & Vogel lab for sharing experiment code, and Jiwoong Park for sharing a movie annotation program. We thank Yuan Chang Leong, Seok-Jun Hong, Emily S Finn, and JeongJun Park for helpful discussions and comments on the manuscript. We thank many researchers who open-sourced their data and code that was used in the project. Research was supported by resources provided by the University of Chicago Research Computing Center. Institute for Basic Science Grant IBS R015-D1 (WMS) National Research Foundation of Korea NRF-2019M3E5D2A01060299, and NRF-2019R1A2C1085566 (WMS) National Science Foundation BCS-2043740 (MDR).

## Additional information

### Funding

| Funder | Grant reference number | Author |
|---|---|---|
| Institute for Basic Science | R015-D1 | Won Mok Shim |
| National Research Foundation of Korea | NRF-2019M3E5D2A01060299 | Won Mok Shim |
| National Research Foundation of Korea | NRF-2019R1A2C1085566 | Won Mok Shim |
| National Science Foundation | BCS-2043740 | Monica D Rosenberg |

| Funder | Grant reference number | Author |
|--------|------------------------|--------|

The funders had no role in study design, data collection and interpretation, or the decision to submit the work for publication.

## Author contributions

Hayoung Song, Conceptualization, Data curation, Formal analysis, Validation, Investigation, Visualization, Methodology, Writing – original draft, Project administration, Writing – review and editing; Won Mok Shim, Conceptualization, Supervision, Funding acquisition, Validation, Investigation, Methodology, Project administration, Writing – review and editing; Monica D Rosenberg, Conceptualization, Supervision, Funding acquisition, Validation, Investigation, Methodology, Writing – original draft, Project administration, Writing – review and editing

## Author ORCIDs

Hayoung Song  http://orcid.org/0000-0002-5970-8076
Monica D Rosenberg  http://orcid.org/0000-0001-6179-4025

## Ethics

Human subjects: Informed consent and consent to publish were obtained from the participants prior to the experiments, and the possible consequences of the study were explained. The study was approved by the Institutional Review Board of Sungkyunkwan University.

## Decision letter and Author response

Decision letter https://doi.org/10.7554/eLife.85487.sa1
Author response https://doi.org/10.7554/eLife.85487.sa2

# Additional files

## Supplementary files

• MDAR checklist

## Data availability

Raw fMRI data from the SitcOm, Nature documentary, Gradual-onset continuous performance task (SONG) dataset are available on OpenNeuro; https://openneuro.org/datasets/ds004592/versions/1.0.1. Behavioral data, processed fMRI output, and main analysis scripts are published on Github (copy archived at *Song, 2023*).

The following dataset was generated:

| Author(s) | Year | Dataset title | Dataset URL | Database and Identifier |
|-----------|------|---------------|-------------|-------------------------|
| Song H, Shim WM, Rosenberg MD | 2023 | SONG dataset | https://doi.org/10.18112/openneuro.ds004592.v1.0.1 | OpenNeuro, 10.18112/openneuro.ds004592.v1.0.1 |

The following previously published datasets were used:

| Author(s) | Year | Dataset title | Dataset URL | Database and Identifier |
|---|---|---|---|---|
| Nastase SA, Liu Y, Hillman H, Zadbood A, Hasenfratz L, Keshavarzian N, Chen J, Honey CJ, Yeshurun Y, Regev M, Nguyen M, Chang CHC, Baldassano C, Lositsky O, Simony E, Chow MA, Leong YC, Brooks PP, Micciche E, Choe G, Goldstein A, Vanderwal T, Halchenko YO, Norman KA, Hasson U | 2020 | Narratives: fMRI data for evaluating models of naturalistic language comprehension | https://doi.org/10.18112/openneuro.ds002345.v1.1.4 | OpenNeuro, 10.18112/openneuro.ds002345.v1.1.4 |
| Chen J, Leong YC, Honey CJ, Yong CH, Norman KA, Hasson U | 2018 | Sherlock | https://doi.org/10.18112/openneuro.ds001132.v1.0.0 | OpenNeuro, 10.18112/openneuro.ds001132.v1.0.0 |
| Margulies DS, Ghosh SS, Goulas A, Falkiewicz M, Huntenburg JM, Langs G, Bezgin G, Eickhoff SB, Castellanos FX, Petrides M, Jefferies E, Smallwood J | 2016 | Situating the default-mode network along a principal gradient of macroscale cortical organization | https://identifiers.org/neurovault.collection:1598 | NeuroVault, 1598 |

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
