## [Editor Report]

This valuable study examines the distribution of four states of brain activity across a variety of cognitive conditions, linking systems neuroscience with cognition and behavior. The work is convincing, using null models and replication in independent datasets to support their findings.

---

## [Decision Letter]

**Decision letter after peer review:**

Thank you for submitting your article "Large-scale neural dynamics in a shared low-dimensional state space reflect cognitive and attentional dynamics" for consideration by *eLife*. Your article has been reviewed by 3 peer reviewers, including Shella Keilholz as the Reviewing Editor and Reviewer #1, and the evaluation has been overseen by Chris Baker as the Senior Editor. The following individual involved in the review of your submission has agreed to reveal their identity: James M Shine (Reviewer #2).

Essential revisions:

1) Please discuss the results, and the 'base state' in particular, in the context of prior studies to help readers understand how this work fits into the existing literature.

2) Please respond to Rev 2's concern that features of the analysis algorithms render some similarities inevitable.

3) Please respond to Rev 3's comments about model training, the selection of the number of states, null models, and the HMM.

*Reviewer #2 (Recommendations for the authors):*

I have provided some suggestions below that I hope will help to improve the manuscript.

P5L36 – In addition to covariance strength, I'm curious whether the authors considered the variance of the covariance matrix.

P7L13 – the authors state that the overlap between the results of the HMM and the gradient analysis is 'intriguing', however, I wonder whether there is a mathematical equivalence to this result. Could the authors conduct both analyses on surrogate data to determine whether similar results were an inevitable by-product of the algorithmic similarities?

Figure 4 – the label 'Default' transitions was slightly confusing, given the fact that the term 'default' is part of the term 'Default Mode Network'. Is there another term that the authors could use?

Figure 5 – the authors may wish to consider changing the box plots to a measure that more transparently plots the individual data points, such as a raincloud or violin plot

General – it would interesting to hear the authors' thoughts on what kinds of organising principles in the brain could provide the means to flexibly shift between these different low-dimensional modes. We have been testing the hypothesis that different arms of the ascending arousal system play a crucial role in this process (e.g., Munn et al., Nature Communications, 2021), and others have shown that similar links between low-dimensional brain state dynamics and arousal exist in whole-brain neuroimaging (e.g., Zhang et al., NeuroImage, 2022).

General – the 'base state' that the authors' report is reminiscent of the results from Saggar et al. (Nature Communications, 2022). It would be interesting to hear the authors' thoughts on the similarities/differences to the interpretation associated with this approach.

General – I was surprised that the study by Viduarre et al. (PNAS, 2017) was not cited, as it used a similar methodological approach (HMM) on resting-state data to infer the underlying dynamical structure of the data. The results, to my mind, overlap well with what the authors present here, so it should be possible to integrate the similarities and differences.

*Reviewer #3 (Recommendations for the authors):*

– Page 5 L46 why do they use the top-5 components

– Grammatical error = : P11 L41 on the other hand is odd wording when there is nothing in the previous sentence this result is compared against.

– Grammatical error: Page 5 L12: HMM -> The HMM

---

## [Author Response]

Essential revisions:1) Please discuss the results, and the 'base state' in particular, in the context of prior studies to help readers understand how this work fits into the existing literature.

We edited the Discussion so that our work can be situated in the context of prior resting-state fMRI studies. We highlight previous findings of the “base state” as a transitional hub. We also discuss work that related arousal fluctuations to brain states, and work that used different clustering or dimensionality reduction algorithms to capture a common set of brain states. By addressing past and recent work on human systems neuroscience, we better situate our findings in the literature and clarify what our work adds: insights into how brain states reflect cognitive and attentional dynamics in diverse contexts.

2) Please respond to Rev 2's concern that features of the analysis algorithms render some similarities inevitable.

In response to Reviewer 2, we conducted additional analyses to ask whether the HMM and functional connectivity gradients share algorithmic similarities that would make the alignment of states to gradient axes inevitable. A shared null model was created for the two algorithms, which allowed us to directly compare the two. By comparing data-specific gradients to data-specific latent states, we show that latent states fall at more extreme positions on the low-dimensional gradient axes than expected by chance. By comparing data-specific gradients to predefined gradients from resting-state fMRI, we demonstrate that a low-dimensional manifold of neural dynamics is shared across contexts. We made edits to the relevant Results and Methods sections, and supplementary figures.

3) Please respond to Rev 3's comments about model training, the selection of the number of states, null models, and the HMM.

We responded to Reviewer 3’s comments about the robustness of the HMM, including the selection of the number of states, model validations, explained variance, and reliability.

We provide evidence that the model is reliable, robust to different hyperparameter selections, and generalize to unseen individuals and external datasets. We made edits to the Results, Discussion, and supplementary figures.

Reviewer #2 (Recommendations for the authors):I have provided some suggestions below that I hope will help to improve the manuscript.P5L36 – In addition to covariance strength, I'm curious whether the authors considered the variance of the covariance matrix.

Covariance strength computes the sum of the absolute covariance weights of the edges. For the four states defined here, the covariance matrices included both positive and negative edges (Figure 1B). We expected that if the absolute covariance weights of the edges were high, the variance would be high as well. As we expected, the variance of covariance matrices scaled with the covariance strength, such that the values followed the order of the SM state being the highest, DMN and DAN states being comparable, and the base state being the lowest (SM > DMN = DAN > base states). The variance of the across-network covariance values (lower half of the triangle, without values in the diagonal) was: DMN: 0.0261, DAN: 0.0264, SM: 0.0495, base: 0.0138. Because this result mirrored that of the covariance strength measures (SM > DMN = DAN > base states), we did not include this additional information in the manuscript but are happy to add it at the reviewer's or editor’s request.

P7L13 – the authors state that the overlap between the results of the HMM and the gradient analysis is 'intriguing', however, I wonder whether there is a mathematical equivalence to this result. Could the authors conduct both analyses on surrogate data to determine whether similar results were an inevitable by-product of the algorithmic similarities?

We appreciate this important question—whether the alignment between the HMM and gradients is an inevitable byproduct due to their algorithmic similarities. With our original analysis, we could not test this because, as the reviewer pointed out, the null model was created only with the HMM but not with gradients (Author response image 1). That is, the predefined gradients were held intact while we conducted null HMMs on the circular-shifted 25 parcel time series. To test for a possibility of the null HMM states aligning to null gradients due to algorithmic similarities, we created a shared null model for both the HMM and gradients, as was suggested by the reviewer.

To create a shared null model, we needed to estimate gradients specific to our SONG dataset. To directly calculate gradients from the SONG data time series, we (1) extracted the BOLD time series of all voxels, (2) applied 1054 ROI parcellation (1000 cortical parcels of the Schaefer et al. (2018) atlas and 54 subcortical parcels of the Tian et al. (2020) atlas) to extract the mean time series of each parcel, (3) calculated ROI-by-ROI functional connectivity matrix per participant, which was averaged across all participants, (4) and applied the same diffusion embedding algorithm as in Margulies et al. (2016). This resulted in the SONG data-specific gradients (Author response image 1). The second SONG gradient was similar to the first predefined gradient (*r* = 0.876) and the first SONG gradient was similar to the second predefined gradient (*r* = 0.877). Importantly, to generate null distributions, we circular-shifted all voxel time series separately to which we applied the 1054 ROI and 25 ROI parcellations (1,000 iterations). The *z*-normalized time series of the 1054 parcels were used to create null gradients. The corresponding *z*-normalized time series of the 25 parcels were used to create null HMM latent states. This allowed us to directly compare the null models of the gradients and HMMs. Calculating gradients on the null time series resulted in extracting non-meaningful gradients. Projecting the HMM states on the non-meaningful axes resulted in the four states being located near zero for all null iterations, which created sharp null distributions of the latent state positions centered around zero (Author response image 1).

The positions of the HMM states on the actual gradients were significantly at more extreme positions compared to the positions of the null HMM states on the null gradients. The results suggest the correspondence between the HMM and gradient results is not inevitable. However, these results also suggest that projecting HMM states on non-meaningful gradient axes is almost guaranteed to converge to zero. That is, the HMM states tile the gradients of brain connectome only when the fMRI time series meaningfully covary over time. Thus, our response to reviewer is that, even with the shared null model, the possibility of “inevitable alignment” of the two analyses cannot be directly ruled out because aligning HMM states to non-meaningful axes of the gradient space (due to disrupted functional connectivity matrix) becomes non-meaningful.

**Author response image 1. sa2fig1:** Test for the alignment between the latent states and the predefined and data-specific gradients. (**A**) (Left) Visualization of the top 2 gradients of [70]. (Right) Colored lines show the positions of the four latent states of the SONG dataset situated on the first and second gradient axes defined in [70]. Four distributions (shown in overlap) indicate null hidden Markov model (HMM) states situated on the Margulies et al.'s (2016) gradient axes. Null states were generated by applying the HMM on the circular-shifted time series (1000 iterations; asterisks indicate FDR-p<0.01 compared to respective null distribution). (**B**) (Left) Visualization of the top 2 gradients of the SONG dataset, estimated based on the 1054 parcel time series. (Middle) Visualization of the example top 2 null gradients using 1054 parcels. Voxel time series were circular-shifted and then averaged per 1054 parcel to be used in null gradient estimation. (Right) Colored lines show the four states (HMM on the 25 parcel time series of the SONG dataset) situated on the top 2 gradient axes (gradients estimated from the 1054 parcel time series of the SONG dataset). A shared null model was generated by circular-shifting the voxel time series, which were averaged using 25 parcels for the latent state estimation or 1054 parcels for the gradient estimation. (**C**) (Left) Visualization of the top 2 gradients of the SONG dataset, estimated based on the 25 parcel time series also used in the HMM. (Middle) Visualization of the example top 2 null gradients using 25 parcels. Circular-shifted 25 parcel time series, used to estimate null HMM states, were also used to estimate functional connectivity gradients. (Right) Colored lines show the four states (HMM on the 25 parcel time series of the SONG dataset) situated on the top 2 gradient axes (gradients estimated from the 25 parcel time series of the SONG dataset). A shared null model was generated by circular-shifting 25 parcel time series.

Our analysis depicted in Author response image 1 entailed one potential limitation. When circular-shifting voxel time series to generate a null model, similarity structure between nearby voxels (inherent spatial smoothness) are disrupted. Thus, the averaged voxel time series for each ROI may not have similar properties (e.g., in frequency or amplitude) as the actual ROI time series.

Thus, we performed a secondary analysis in which we compared the SONG gradients and SONG HMM states, both using 25 parcels (Author response image 1). Null distributions were created by circular-shifting parcel time series rather than voxel time series as before. This retains temporal properties of fMRI time series of each parcel while disrupting covariance structure across parcels. Here, because the 25 x 25 functional connectivity matrix was already small, we did not threshold functional connectivity matrix at 90% (row-wise) for sparsity. Instead, all negative values in the functional connectivity matrix were zeroed prior to computing the affinity matrix.

Surprisingly, even with a small number of parcels, the top 2 gradients were replicated (Author response image 1). Again, the second SONG gradient was similar to the first predefined gradient (*r* = 0.845) and the first SONG gradient was similar to the second predefined gradient (*r* = 0.832). Gradient embedding similarity was calculated with Pearson’s correlations of the 25 parcel patterns. Replicating our findings with the 1054 parcels, the null HMM states situated on the null gradients converged to near zero. The actual HMM states fell at more extreme positions on the actual gradient axes than would be expected by chance. The first gradient dissociated the SM state from the DMN and DAN states, the second gradient dissociated the DMN state from the DAN and SM states, and the base state was consistently situated near zero.

Together, by creating a shared null model for HMM and gradient analyses, we partially addressed the reviewer’s question. Given that the positions of the HMM states on the actual gradients were at more extreme positions compared to chance, we demonstrated robustness and statistical significance of our findings. However, when gradients themselves were disrupted, aligning the HMM states to these non-meaningful gradients also became non-meaningful, thus converging to zero. Therefore, we demonstrated that states do not always fall at the extremes of gradient axes but could not directly test for mathematical equivalence between two algorithms using the null model.

While we could not assess mathematical equivalence between the algorithms, we note that HMM is applied to fMRI time series to infer latent state dynamics whereas gradient analysis is applied to static functional connectivity (which collapses variance over time) to capture low-dimensional axes that explain the most variances in the functional connectome. Given these differences, we found it surprising that the HMM states tile the gradient space defined from a different dataset. As proposed by Brown et al. (2021), this evidence supports that the functional architecture (gradients) constrains the range of brain activity dynamics (latent state dynamics).

These analyses and results were added to the Results, Methods, and Figure 1—figure supplement 8. Because Reviewer 3 also raised a concern that the principal component analysis is not a fair comparison to the predefined gradients, we replaced the PCA results with the dataspecific gradient results in the revised manuscript. Here, we attach changes to the Results section and the addition of Figure 1—figure supplement 8.

Manuscript, page 5-6

“We hypothesized that the spatial gradients reported by Margulies et al. (2016) act as a lowdimensional manifold over which large-scale dynamics operate (Bolt et al., 2022; Brown et al., 2021; Karapanagiotidis et al., 2020; Turnbull et al., 2020), such that traversals within this manifold explain large variance in neural dynamics and, consequently, cognition and behavior (Figure 1C). To test this idea, we situated the mean activity values of the four latent states along the gradients defined by Margulies et al. (2016) (see *Methods*). The brain states tiled the two-dimensional gradient space with the base state at the center (Figure 1D; Figure1—figure supplement 7). The Euclidean distances between these four states were maximized in the two-dimensional gradient space, compared to a chance where the four states were inferred from circular-shifted time series (*p* < 0.001). For the SONG dataset, the DMN and SM states fell at more extreme positions of the primary gradient than expected by chance (both FDR-*p* values = 0.004; DAN and SM states, FDR*p* values = 0.171). For the HCP dataset, the DMN and DAN states fell at more extreme positions on the primary gradient (both FDR-*p* values = 0.004; SM and base states, FDR-*p* values = 0.076). No state was consistently found at the extremes of the secondary gradient (all FDR-*p* values > 0.021).

We asked whether the predefined gradients explain as much variance in neural dynamics as latent subspace optimized for the SONG dataset. To do so, we applied the same nonlinear dimensionality reduction algorithm to the SONG dataset’s ROI time series. Of note, the SONG dataset includes 18.95% rest, 15.07% task, and 65.98% movie-watching data whereas the data used by Margulies et al. (2016) was 100% rest. Despite these differences, the SONG-specific gradients closely resembled the predefined gradients, with significant Pearson’s correlations observed for the first (*r* = 0.876) and second (*r* = 0.877) gradient embeddings (Figure 1—figure supplement 8). Gradients identified with the HCP data also recapitulated Margulies et al.’s (2016) first (*r* = 0.880) and second (*r* = 0.871) gradients. We restricted our analysis to the first two gradients because the two gradients together explained roughly 50% of the entire variance of functional brain connectome (SONG: 46.94%, HCP: 52.08%), and the explained variance dropped drastically from the third gradients (more than 1/3 drop compared to the second gradients). The degrees to which the first two predefined gradients explained whole-brain fMRI time series (SONG: *r*^2^ = 0.097, HCP: 0.084) were comparable to the amount of variance explained by the first two data-specific gradients (SONG: *r*^2^ = 0.100, HCP: 0.086; Figure 1—figure supplement 8). Thus, the low-dimensional manifold captured by Margulies et al. (2016) gradients is highly replicable, explaining brain activity dynamics as well as data-specific gradients, and is largely shared across contexts and datasets. This suggests that the state space of whole-brain *dynamics* closely recapitulates low-dimensional gradients of the *static* functional brain connectome.”

“Figure 1—figure supplement 8.

Comparisons between predefined and data-specific gradients. (A) Visualization of the Margulies et al.’s (2016) first two gradient embeddings (left). The predefined gradients were downloaded from https://identifiers.org/neurovault.collection:1598. Visualization of the first two gradient embeddings defined from the SONG data (middle) and the HCP data (right). The gradients were computed from the mean 1054 ROI-by-ROI functional connectivity matrix of all participants in each dataset. The 1054 parcels include 1000 cortical ROIs of the Schaefer et al. (2018) atlas and 54 subcortical ROIs of Tian et al. (2020) atlas. (B) Pearson’s correlations between the top-5 Margulies et al. (2016) gradient embeddings and the SONG data (top) and HCP data (bottom) gradient embeddings. The gradient embeddings were estimated from 1054 ROIs. (C) Variance of the SONG data participants’ average functional connectivity explained by the SONG-specific gradients (solid line). Variance of the HCP data participants’ average functional connectivity explained by the HCP-specific gradients (dashed line). (D) (Left) Variance of the SONG data fMRI time series (1054 ROIs) explained by the first two SONG data gradients (black) and Margulies et al. (2016) gradients (gray). (Right) Variance of the HCP data fMRI time series explained by the first two HCP data gradients (black) and Margulies et al. (2016) gradients (gray). The explained variance was calculated by the mean of squared Pearson’s correlations (**r^2^**) between the 1054 ROI fMRI time series and the gradient-projected time series.”

Figure 4 – the label 'Default' transitions was slightly confusing, given the fact that the term 'default' is part of the term 'Default Mode Network'. Is there another term that the authors could use?

Thank you for pointing this out. We changed “Default transitions to the DMN state” to “Typical transitions to the DMN state” in Figure 4.

Figure 5 – the authors may wish to consider changing the box plots to a measure that more transparently plots the individual data points, such as a raincloud or violin plot

We agree with the reviewer’s suggestion about showing individual data points. We tried using the raincloud or violin plots, but since there were so many graphs, the different shapes of the violin plots looked busy. To achieve both simplicity and transparency, we overlaid individual data points on our original bar graphs. We changed Figure 5 and the legend accordingly. For consistency, all main-figure bar graphs now show individual data points.

General – it would interesting to hear the authors' thoughts on what kinds of organising principles in the brain could provide the means to flexibly shift between these different low-dimensional modes. We have been testing the hypothesis that different arms of the ascending arousal system play a crucial role in this process (e.g., Munn et al., Nature Communications, 2021), and others have shown that similar links between low-dimensional brain state dynamics and arousal exist in whole-brain neuroimaging (e.g., Zhang et al., NeuroImage, 2022).

As with the reviewer, we also wondered what drives flexible, macroscopic changes in neural states. We will share a story of how our thoughts and analyses progressed as we conducted the study.

After conducting the HMM (Figure 1), we were curious about what causes the transient change in the metastable brain states (i.e., transition to a different HMM state in the latent state sequence). Our initial hypothesis was that a *selective* set of cortico-subcortical interactions would drive global neural state transitions. This hypothesis was motivated by Munn et al. (2021) and Shine (2019), which argued that the neuromodulatory signals from the ascending arousal system, passing the thalamus, alter the low-dimensional energy landscape of cortical dynamics. We were also inspired by Greene et al. (2020), which categorized zones of functional network integration and specificity in the subcortex from its interactions with the cortical networks.

Because our imaging protocol did not allow us to directly investigate signals from smaller deep brain regions such as locus coeruleus as in Munn et al. (2021), we instead investigated thalamus activity and its interaction with large-scale cortical networks. We hypothesized that the higher-order area of the thalamus may trigger neural state transitions, in which case we expected a phasic increase in this region’s cofluctuation with cortical network regions.

**Author response image 2. sa2fig2:** Schematics of the initial hypothesis and the observed result. (**A**) Schematics of the initial hypothesis. We hypothesized that neural state transitions would be triggered by the transient increase in cofluctuation of selective pairs of brain regions. (**B**) Schematics of the observed results (Figure 2). We observed a global decrease in cofluctuation prior to state transitions.

To test this hypothesis—that a phasic increase in corticothalamic cofluctuations would occur prior to HMM state transitions (Author response image 2)—we extracted time series of 16 subareas of the thalamus and 17 Yeo et al. (2011) cortical networks. We calculated the cofluctuation time series of the 272 corticothalamic pairs. We then applied a new HMM to the time series of the 17 cortical networks only (instead of the 25 cortical and subcortical parcels we used in our main manuscript analysis). The four states inferred by this cortex-only HMM were similar to the results with the 25 parcels, with the mean activity pattern correlation *r*=0.988 for the DMN state, *r*=0.999 for the DAN state, *r*=0.989 for the SM state, and *r*=0.985 for the base state.

We then extracted cofluctuation values at each time point between [*t-3*, *t*+3] centered on new state onsets at time *t*. We averaged these values across every state transition for each participant. For every fMRI run, we tested whether any of the corticothalamic pairs consistently increased their cofluctuation prior to state transitions (i.e., higher cofluctuation values at time *t*-1, compared to *t*+3, compared using the Wilcoxon signed-rank test, *p* < 0.01). Across all runs, *none* of the pairs exhibited increased cofluctuation prior to state transitions. On the contrary, *most* of the pairs exhibited significant decreases in cofluctuation at time *t*-1, as observed in Figure 2 in the manuscript and schematized in Author response image 2. Thus, instead of a transient increase in cofluctuation of corticothalamic edges, a transient and global decrease in cofluctuation better explained neural state transitions detected by the HMM.

Another possibility we considered was that the thalamus received indirect signals from the brainstem and basal forebrain, as these regions show phasic bursts in activity (Munn et al., 2021). If this were the case, we predicted that thalamic activity would increase 200-300ms after signal bursts in the deep brain regions (Shine, 2019).

To test this second hypothesis, we time-aligned 16 thalamic regions’ BOLD activity to [*t-3*, *t*+3] time windows centered on the new state onsets at *t*. When we compared the averaged time-aligned BOLD activity (*z*) to the null distribution where we circular-shifted the HMM latent state sequence, none of the 16 thalamic regions showed significant differences in BOLD activity compared to chance. Our analysis did not detect modulation of thalamic BOLD activity that was aligned to HMM state transitions.

Because of our methodological limitation in probing deeper brain area activity and lack of support for the hypothesis that corticothalamic interactions drive state transitions, we did not make strong claims about the role of the ascending arousal system in our manuscript. Nonetheless, we strongly agree that continuing to explore what causes macroscale state transitions will provide significant insights into cognitive, attentional, and arousal dynamics and is an important direction for future work. We further hope our reports on global desynchronization preceding brain state transitions may inspire future research.

General – the 'base state' that the authors' report is reminiscent of the results from Saggar et al. (Nature Communications, 2022). It would be interesting to hear the authors' thoughts on the similarities/differences to the interpretation associated with this approach.

We greatly appreciate the reviewer’s suggestion of prior papers. We were not aware of previous findings of the base state at the time of writing the manuscript, so it was reassuring to see consistent findings. In the Discussion, we highlighted the findings of Chen et al. (2016) and Saggar et al. (2022). Both these studies highlighted the role of the base state as a “hub”-like transition state. However, as the reviewer noted, these studies did not address the functional relevance of this state to cognitive states because both were based on resting-state fMRI.

In our revised Discussion, we write that our work *replicates* previous findings of the base state that consistently acted as a transitional hub state in macroscopic brain dynamics. We also note that our project expands this line of work by characterizing what *functional roles* the base state plays in multiple contexts: The base state indicated high attentional engagement and exhibited the highest occurrence proportion as well as longest dwell times during naturalistic movie watching. The base state’s functional involvement was comparatively minor during controlled tasks.

Manuscript, page 17-18

“Past resting-state fMRI studies have reported the existence of the base state. Chen et al. (2016) used the HMM to detect a state that had ‘less apparent activation or deactivation patterns in known networks compared with other states.’ This state had the highest occurrence probability among the inferred latent states, was consistently detected by the model, and was most likely to transition to and from other states, all of which mirror our findings here. The authors interpret this state as an ‘intermediate transient state that appears when the brain is switching between other more reproducible brain states.’ The observation of the base state was not confined to studies using HMMs. Saggar et al. (2022) used topological data analysis to represent a low-dimensional manifold of resting-state whole-brain dynamics as a graph, where each node corresponds to brain activity patterns of a cluster of time points. Topologically focal ‘hub’ nodes were represented uniformly by all functional networks, meaning that no characteristic activation above or below the mean was detected, similar to what we observe with the base state. The transition probability from other states to the hub state was the highest, demonstrating its role as a putative transition state.

However, the functional relevance of the base state to human cognition had not been explored previously. We propose that the base state, a transitional hub (Figure 2B) positioned at the center of the gradient subspace (Figure 1D), functions as a state of natural equilibrium. Transitioning to the DMN, DAN, or SM states reflects incursion away from natural equilibrium (Deco et al., 2017; Gu et al., 2015), as the brain enters a functionally modular state. Notably, the base state indicated high attentional engagement (Figure 5E and F) and exhibited the highest occurrence proportion (Figure 3B) as well as the longest dwell times (Figure 3—figure supplement 1) during naturalistic movie watching, whereas its functional involvement was comparatively minor during controlled tasks. This significant relevance to behavior verifies that the base state cannot simply be a byproduct of the model. We speculate that susceptibility to both external and internal information is maximized in the base state—allowing for roughly equal weighting of both sides so that they can be integrated to form a coherent representation of the world—at the expense of the stability of a certain functional network (Cocchi et al., 2017; Fagerholm et al., 2015). When processing rich narratives, particularly when a person is fully immersed without having to exert cognitive effort, a less modular state with high degrees of freedom to reach other states may be more likely to be involved. The role of the base state should be further investigated in future studies.”

General – I was surprised that the study by Viduarre et al. (PNAS, 2017) was not cited, as it used a similar methodological approach (HMM) on resting-state data to infer the underlying dynamical structure of the data. The results, to my mind, overlap well with what the authors present here, so it should be possible to integrate the similarities and differences.

Yes, Vidaurre et al. (Vidaurre et al., 2018, 2017) was the primary paper that motivated our HMM analysis. The paper was cited in the original manuscript. We agree with the reviewer about noting the similarities and differences of our results from this paper, and have added this to the Discussion.

Manuscript, page 15-16

“This perspective is supported by previous work that has used different methods to capture recurring low-dimensional states from spontaneous fMRI activity during rest. For example, to extract time-averaged latent states, early resting-state analyses identified task-positive and tasknegative networks using seed-based correlation (Fox et al., 2005). Dimensionality reduction algorithms such as independent component analysis (Smith et al., 2009) extracted latent components that explain the largest variance in fMRI time series. Other lines of work used timeresolved analyses to capture latent state dynamics. For example, variants of clustering algorithms, such as co-activation patterns (Liu et al., 2018; Liu and Duyn, 2013), k-means clustering (Allen et al., 2014), and HMM (Baker et al., 2014; Chen et al., 2016; Vidaurre et al., 2018, 2017), characterized fMRI time series as recurrences of and transitions between a small number of states. Time-lag analysis was used to identify quasiperiodic spatiotemporal patterns of propagating brain activity (Abbas et al., 2019; Yousefi and Keilholz, 2021). A recent study extensively compared these different algorithms and showed that they all report qualitatively similar latent states or components when applied to fMRI data (Bolt et al., 2022). While these studies used different algorithms to probe data-specific brain states, this work and ours report common latent axes that follow a long-standing theory of large-scale human functional systems (Mesulam, 1998). Neural dynamics span principal axes that dissociate unimodal to transmodal and sensory to motor information processing systems.”

Reviewer #3 (Recommendations for the authors):– Page 5 L46 why do they use the top-5 components

Thank you for raising this point. In our revised manuscript, we only use the top 2 gradients after assessing the variance that each gradient explains functional brain connectivity (Figure 1—figure supplement 8). Because the top 2 gradients explained almost half of the variance of functional connectivity, we considered that the first two gradients capture meaningful lowdimensional manifold of functional brain organization.

Manuscript, page 6

“We restricted our analysis to the first two gradients because the two gradients together explained roughly 50% of the variance of the functional brain connectome (SONG: 46.94%, HCP: 52.08%), and the explained variance dropped drastically from the third gradients (more than 1/3 drop compared to the second gradients).”

– Grammatical error = : P11 L41 on the other hand is odd wording when there is nothing in the previous sentence this result is compared against.

We used the term “on the other hand” to emphasize that paying attention to a comedy sitcom involves *different* cognitive processes than attending to controlled psychological tasks.

– Grammatical error: Page 5 L12: HMM -> The HMM

Thank you, we fixed the error according to the reviewer’s suggestion.